# X-EviProbe: Post-hoc Parameter-Free Evidential Uncertainty Quantification for Frozen Graph Neural Networks

Chenghua Guo [1]   Sihong Xie [2]   Xi Zhang [1]

## Abstract

Reliable uncertainty quantification (UQ) is crucial for deploying graph neural networks (GNNs) in safety-critical settings, yet dominant solutions either rely on costly multi-pass sampling or require retraining—often using *black-box auxiliary* models—to obtain evidential semantics. We propose **X-EviProbe**, a simple and parameter-free *post-hoc* framework that turns a *frozen* GNN into an evidential predictor with a decomposable view of epistemic vs. aleatoric uncertainty. X-EviProbe constructs class-wise Dirichlet evidence by probing the frozen latent space and the model's native outputs, and incorporates graph structure via lightweight evidence-strength propagation. This yields a transparent evidential representation without retraining or additional neural components. Extensive experiments on seven benchmarks show that X-EviProbe consistently ranks among the top methods for both OOD detection and misclassification detection, improving AUROC by up to **33.4%** and **8.7%** over the strongest baselines.

## 1. Introduction

Graph Neural Networks (GNNs) model complex dependencies and topological structures (Kipf & Welling, 2017; Velickovic et al., 2018) and have been widely applied in domains such as drug discovery (Stokes et al., 2020), intrusion detection (Zhong et al., 2024), fraud detection (Liu et al., 2021; 2020b), and medical diagnosis (Ahmedt-Aristizabal et al., 2021; Li et al., 2022). However, standard GNNs typically output deterministic point estimates (e.g., softmax probabilities) and can be overconfident even when

wrong (Guo et al., 2017), which is problematic in safety-critical deployments (Amodei et al., 2016).

Several methods have been explored for GNN uncertainty quantification. Sampling-based approaches (e.g., Bayesian GNNs (Zhang et al., 2019; Pal et al., 2019), Deep Ensembles (Lakshminarayanan et al., 2017), and SDE-based methods (Bergna et al., 2025)) measure predictive variability across stochastic passes. Deterministic approaches, most notably Evidential Deep Learning (EDL), construct a Dirichlet distribution to represent evidence magnitude and its class-wise allocation, enabling single-pass decomposition into aleatoric and epistemic uncertainty (Sensoy et al., 2018).

Despite these advancements, it remains difficult to obtain a *post-hoc, principled, and decomposable* evidential view of uncertainty for a *frozen* GNN. Variance-based scores are hard to interpret as concrete evidence, while standard EDL (Zhao et al., 2020) is typically not post-hoc and requires retraining. On graphs, where pretrained backbones are often deployed as-is, plug-and-play evidential methods (Yu et al., 2025) rely on auxiliary models and add opacity. We therefore seek a transparent evidential construction that operates directly on frozen representations and outputs, and supports epistemic/aleatoric decomposition.

In this work, we show that post-hoc evidential uncertainty can be revealed from a converged, frozen GNN without auxiliary learning. Our perspective follows two principles: (i) under mild regularity conditions (e.g., Lipschitz continuity), larger distance in frozen latent space from a *reference set* indicates weaker latent support and higher support-induced epistemic uncertainty; (ii) after convergence, residual errors are expected to be enriched near overlap/low-margin regions (cf. *Neural Collapse* (Papyan et al., 2020)), so proximity to misclassified reference nodes indicates aleatoric ambiguity.

The challenge is to decode these latent cues and fuse them with native outputs into a coherent evidential representation. We propose **X-EviProbe**[1], a parameter-free post-hoc framework that constructs a Dirichlet distribution by integrating geometry as a reliability gate, logit magnitude as evidence strength, and predicted probability as class-wise evidence

---

[1]Key Laboratory of Trustworthy Distributed Computing and Service (MoE), Beijing University of Posts and Telecommunications, China [2]Thrust of Artificial Intelligence, The Hong Kong University of Science and Technology (Guangzhou), China. Correspondence to: Xi Zhang <zhangx@bupt.edu.cn>.

*Proceedings of the 43rd International Conference on Machine Learning*, Seoul, South Korea. PMLR 306, 2026. Copyright 2026 by the author(s).

---

[1]Code is available at: https://github.com/chenghuaguo/X-EviProbe.

| Method | Single pass | No black-box aux model | Class-wise evidence | Model req.: only Asm. 2.1 | Example |
|---|---|---|---|---|---|
| Sampling (Bayes/SDE/Ens.) | × | ✓ | × | × | (Zhang et al., 2019; Bergna et al., 2025) (Lakshminarayanan et al., 2017) |
| Standard EDL | ✓ | ✓ | ✓ | × | (Sensoy et al., 2018; Zhao et al., 2020) |
| Plug-in EDL | ✓ | × | ✓ | ✓ | (Yu et al., 2025) |
| **X-EviProbe** | ✓ | ✓ | ✓ | ✓ | (Ours) |

*Table 1.* Capability comparison of representative GNN uncertainty-quantification paradigms. We highlight *class-wise evidential semantics* (Dirichlet evidence per class). X-EviProbe constructs class-wise evidence from a pretrained GNN via post-hoc probing, and avoids multi-pass sampling and black-box auxiliary models while requiring only Assumption 2.1 on the model form.

allocation. Table 1 summarizes the targeted capabilities.

To summarize, our main contributions are as follows:

- **Methodologically**, we propose **X-EviProbe**, a parameter-free *post-hoc* framework that turns a converged GNN into an evidential predictor with a decomposable view of epistemic vs. aleatoric uncertainty. It operates directly on the frozen latent geometry and native model outputs, incorporates graph structure via light propagation of a *scalar evidence strength*, and requires no retraining or auxiliary networks.

- **Theoretically**, we provide analysis to motivate the key design choices in X-EviProbe (Sec. 4). In particular, we connect the $L_1$ nearest-neighbor distance to a *Lipschitz-motivated* excess-risk bound for support-induced epistemic uncertainty, and interpret misclassified reference samples (i.e., "error anchors") as sparse indicators of high aleatoric ambiguity; we further clarify why the geometry–logits–probability fusion yields a coherent evidential construction.

- **Experimentally**, X-EviProbe achieves top-tier performance on both OOD detection and misclassification detection across seven benchmarks (Sec. 5). It consistently ranks among the best methods and improves AUROC by up to **33.4%** (OOD) and **8.7%** (misclassification) over the strongest baselines. Complementary ablations verify the importance of our key design choices (anchor definition, Laplace/$L_1$ geometry, reference-set composition, and individual modules), showing that the geometric/structural components are critical, while probability allocation and logit magnitude exhibit stream- and task-specific effects.

## 2. Preliminaries

In this section, we establish the problem setting for uncertainty quantification within the context of semi-supervised node classification. We then review the theoretical foundations of Evidential Deep Learning (EDL) (Sensoy et al., 2018) and graph propagation mechanisms, such as Person-

alized PageRank (PPR) (Klicpera et al., 2019), which serve as the building blocks for our framework.

### 2.1. Notations and Problem Definition

Let $\mathcal{G} = (\mathcal{V}, \mathbf{A}, \mathbf{X})$ be an attributed graph with $N$ nodes, where $\mathbf{A} \in \mathbb{R}^{N \times N}$ is the adjacency matrix and $\mathbf{X} \in \mathbb{R}^{N \times F}$ denotes the node features. The degree matrix is $\mathbf{D} = \text{diag}(d_1, \ldots, d_N)$. We consider a semi-supervised setting where a subset of nodes $\mathcal{V}_L \subset \mathcal{V}$ have labels $\mathbf{Y}_L \in \{0, 1\}^{|\mathcal{V}_L| \times K}$ (one-hot encoded for $K$ classes).

**Problem Formulation.** Given a pre-trained, frozen GNN classifier $f$, our goal is to perform *Post-hoc Uncertainty Quantification* for unlabeled nodes $\mathcal{V}_U$. Specifically, for any query node $i$, we aim to:

1. **Construct** a Dirichlet distribution parameterized by $\boldsymbol{\alpha}_i$ to characterize the predictive belief. Crucially, this construction leverages the *intrinsic geometry* of the frozen latent space in conjunction with the model's outputs (i.e., unnormalized scores and probabilities) to calibrate the uncertainty estimates.

2. **Decompose** the total uncertainty estimate into distinct aleatoric and epistemic components without the need to retrain the backbone model.

### 2.2. Evidential Uncertainty Theory

Standard classifiers provide point estimates via Softmax. In contrast, Evidential Deep Learning (EDL) (Sensoy et al., 2018) models the classification probability as a random variable following a Dirichlet distribution $\text{Dir}(\mathbf{p}_i | \boldsymbol{\alpha}_i)$, where $\boldsymbol{\alpha}_i \in \mathbb{R}_+^K$ are concentration parameters.

From a Subjective Logic perspective, $\boldsymbol{\alpha}_i$ relates to the accumulated evidence $\mathbf{e}_i$ via $\boldsymbol{\alpha}_i = \mathbf{e}_i + \mathbf{1}$, where $\mathbf{1}$ represents a non-informative prior. The total evidence strength is $S_i = \sum_{c=1}^K \alpha_{ic}$. Following established frameworks (Malinin & Gales, 2018; Yu et al., 2025), we categorize predictive uncertainty into two primary types:

- **Aleatoric Uncertainty** ($u_i^{alea}$): Often associated with inherent data complexity, such as class overlap or ir-

reducible ambiguity. It is typically derived from the expectation of the predictive categorical distribution:

$$u_i^{alea} = -\max_c \mathbb{E}[\pi_{ic}] = -\max_c \left( \frac{\alpha_{ic}}{S_i} \right). \quad (1)$$

- **Epistemic Uncertainty** ($u_i^{epi}$): Generally arises from a lack of knowledge, such as the sparsity of observed data or distributional shifts (OOD). It is quantified as the inverse of the total Dirichlet strength:

$$u_i^{epi} = \frac{K}{S_i}. \quad (2)$$

## 2.3. Graph-based Evidence Propagation

A fundamental advantage of graph-based learning is the ability to leverage structural information. Graph propagation mechanisms, including PPR (Klicpera et al., 2019) and finite-step linear smoothing, are widely adopted to distribute signals across the graph topology.

Inspired by evidence propagation in (Stadler et al., 2021) and energy propagation in (Wu et al., 2023), recent strategies employ propagation schemes to enhance uncertainty estimation by leveraging this structural consistency. Formally, let $\xi^{(0)}$ be an initial signal vector. In this work, we consider the following linear smoothing process:

$$\xi^{(k)} = \gamma \xi^{(k-1)} + (1-\gamma)\hat{\mathbf{A}}\xi^{(k-1)}, \quad (3)$$

where $\gamma \in (0,1]$ controls the trade-off between preserving the previous node-wise signal and mixing information from neighboring nodes, and $\hat{\mathbf{A}}$ is the normalized adjacency matrix (e.g., $\mathbf{D}^{-1}\mathbf{A}$). In Section 3, we utilize this mechanism to incorporate structural smoothness into our derived evidence strength.

## 2.4. Model Architecture Decomposition

We assume the pre-trained model $f$ belongs to the family of Message Passing Neural Networks (MPNNs) and adopts a standard architecture structure.

**Assumption 2.1 (Model Architecture Decomposition).** The prediction model $f$ is composed of $f = \varphi \circ l \circ g$, where

- $g : (\mathbf{A}, \mathbf{X}) \mapsto \mathbf{H} \in \mathbb{R}^{N \times d_H}$ is a GNN encoder that maps the whole graph (adjacency and node features) to a node embedding matrix $\mathbf{H}$, whose $i$-th row $\mathbf{h}_i \in \mathbb{R}^{d_H}$ is the latent representation of node $i$.

- $l : \mathbb{R}^{d_H} \to \mathbb{R}^K$ is the node-wise logit map induced by the frozen classifier, mapping the latent representation $\mathbf{h}_i$ to logits $\mathbf{z}_i$.

- $\varphi(\cdot)$ is a normalization function (typically Softmax) converting logits to probabilities $\hat{\mathbf{p}}_i$.

*Remark* 2.2. This decomposition is satisfied by most standard GNN architectures, including GCN, GAT, and GraphSAGE. Our framework relies on the geometry of $\mathcal{H}$ and the native outputs of the frozen classifier, namely logits and probabilities, while treating the internal mechanism of $g$ as a frozen feature extractor.

## 3. The X-EviProbe Framework

We propose **X-EviProbe**, a post-hoc interpretable uncertainty quantification framework designed specifically for frozen, pre-trained GNNs. As illustrated in Figure 1, our method systematically decouples the collection of evidence in the latent space into two distinct streams: one for Epistemic Uncertainty (EU) and one for Aleatoric Uncertainty (AU). The framework operates in three sequential stages: (1) *Geometric Evidence Extraction*, (2) *Holographic Evidence Fusion*, and (3) *Linear Evidence Strength Propagation*.

### 3.1. Geometric Evidence Extraction

We ground our uncertainty estimation in the intrinsic geometry of the frozen embedding space $\mathcal{H}$. Let $\mathbf{h}_i$ denote the frozen embedding of node $i$ produced by the frozen encoder. We define the *Reference Set* $\mathcal{D}_{\text{ref}} = \mathcal{V}_L \cup \mathcal{V}_{\text{val}}$ as the union of labeled training nodes and validation nodes. We employ a Laplace kernel to measure geometric similarity, with bandwidth $\sigma$ set to the median of each reference node's nearest-neighbor $L_1$ distance. Crucially, we identify distinct geometric sources for different uncertainty types:

**1. Epistemic Evidence ($e_i^{\mathbf{epi}}$).** Motivated by the Lipschitz continuity of neural networks, we assume the prediction risk is locally bounded by the distance to the nearest observed sample. Thus, we define the epistemic evidence as:

$$e_i^{\text{epi}} = \exp\left(-\frac{\min_{j \in \mathcal{D}_{\text{ref}}} \|\mathbf{h}_i - \mathbf{h}_j\|_1}{\sigma}\right). \quad (4)$$

This score quantifies the *geometric support* of the region: $e_i^{\text{epi}} \to 0$ implies the node is isolated from known data.

**2. Aleatoric Evidence ($e_i^{\mathbf{alea}}$).** To capture aleatoric uncertainty arising from data complexity (such as class overlap or labeling noise), we explicitly identify the set of *Error Anchors* $\mathcal{M}_{\text{ref}} \subset \mathcal{D}_{\text{ref}}$. This set comprises all nodes in the reference set (training and validation) that are misclassified by the frozen model. We define:

$$e_i^{\text{alea}} = \exp\left(-\frac{\min_{j \in \mathcal{M}_{\text{ref}}} \|\mathbf{h}_i - \mathbf{h}_j\|_1}{\sigma}\right). \quad (5)$$

By convention, the minimum over an empty anchor set is $+\infty$, yielding $e_i^{\text{alea}} = 0$ when no error anchor is observed. A high value of $e_i^{\text{alea}}$ indicates *confusion proximity*—the query node lies where the model often makes errors.

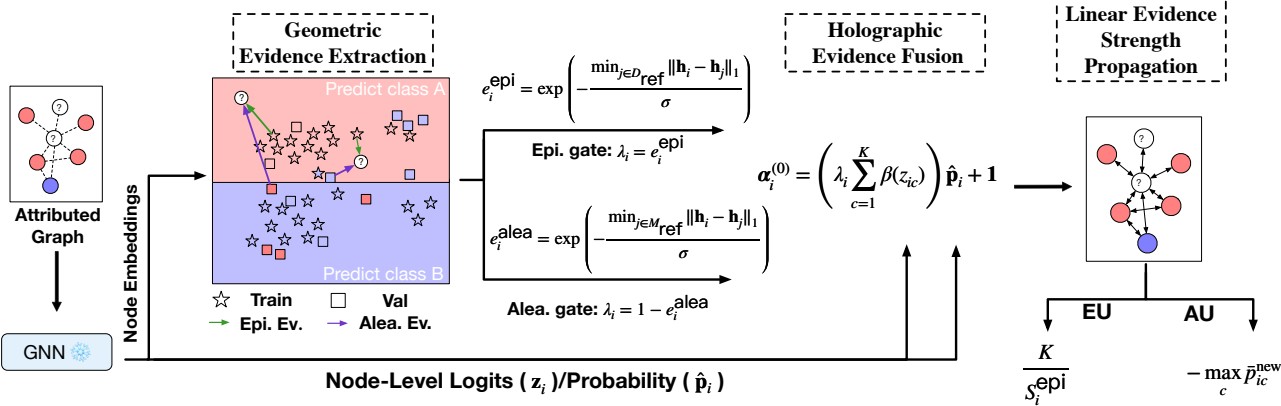

*Figure 1.* Overview of X-EviProbe. We extract geometric evidence in the frozen latent space, fuse it with model outputs to construct Dirichlet evidence, and propagate scalar evidence strength to quantify epistemic and aleatoric uncertainty. Here, $S_i^{\text{epi}}$ denotes the propagated total evidence strength for the epistemic stream, $\bar{p}_{ic}^{\text{new}}$ denotes the rectified posterior probability for class $c$ computed from the aleatoric evidence stream, and $\beta(\cdot)$ denotes the softplus function.

## 3.2. Holographic Evidence Fusion

Unlike prior works that predict Dirichlet parameters $\boldsymbol{\alpha}$ via auxiliary black-box models, we construct them by holographically fusing three sources of information available in the converged model: the latent geometry, the logit magnitude, and the predicted probability. We define a *Geometric Gate*, $\lambda_i$, which modulates the model's confidence based on the geometric evidence extracted above.

- **For Epistemic Uncertainty:** We set $\lambda_i = e_i^{\text{epi}}$. Here, the gate suppresses evidence if the sample is geometrically isolated.

- **For Aleatoric Uncertainty:** We set $\lambda_i = 1 - e_i^{\text{alea}}$. Here, the gate suppresses evidence if the sample is close to known errors (high confusion).

The initial Dirichlet parameters $\boldsymbol{\alpha}_i^{(0)}$ are then computed as:

$$\boldsymbol{\alpha}_i^{(0)} = \underbrace{\left( \lambda_i \sum_{c=1}^{K} \text{softplus}(z_{ic}) \right)}_{\text{Gated Evidence Mass}} \cdot \hat{\mathbf{p}}_i + \mathbf{1}. \qquad (6)$$

This construction embodies a hierarchical intuition of data flow: the *Latent Geometry* ($\lambda_i$) gates the *Logit Magnitude* ($\sum \text{softplus}(\mathbf{z})$), which in turn scales the *Prediction Direction* ($\hat{\mathbf{p}}$). This ensures that the final evidential distribution is consistent with both the model's discriminative power and the geometric reliability of the input.

## 3.3. Linear Evidence Strength Propagation

While the previous step generates node-wise evidence, it does not explicitly propagate uncertainty over the graph topology. We adopt the assumption that uncertainty exhibits *structural smoothness*—neighboring nodes are likely to possess similar levels of predictive reliability or ambiguity (Yu

et al., 2025; Wu et al., 2023). To leverage this without the high computational cost of propagating dense Dirichlet vectors, we propagate the scalar **Total Evidence Strength** $S_i = \sum_{c=1}^{K} \alpha_{ic}$. Let $\mathbf{S}^{(0)} \in \mathbb{R}^N$ be the vector of initial strengths derived from Eq. (6). We utilize the lazy linear smoothing operator defined in Eq. (3):

$$\mathbf{S}^{(k)} = \gamma \mathbf{S}^{(k-1)} + (1-\gamma)\hat{\mathbf{A}}\mathbf{S}^{(k-1)}, \qquad (7)$$

where $\gamma$ is the self-retention coefficient. We set $\gamma = 0.5$ by default, assigning equal weight to the previous propagated evidence strength and the one-hop neighborhood aggregate at each propagation step. A useful by-product of this linear formulation is *traceability*: letting $\mathbf{M}_\gamma = \gamma\mathbf{I} + (1-\gamma)\hat{\mathbf{A}}$, the propagated strength after $T$ steps satisfies $\mathbf{S}^{(T)} = \mathbf{M}_\gamma^T \mathbf{S}^{(0)}$. Hence, the coefficient $[\mathbf{M}_\gamma^T]_{ij}$ gives the per-unit influence weight of node $j$'s initial evidence strength on node $i$'s propagated strength, while $[\mathbf{M}_\gamma^T]_{ij} S_j^{(0)}$ gives its actual contribution. This allows the evidence pattern underlying high uncertainty to be traced back to influential neighbors and their evidence components.

## 3.4. Final Uncertainty Quantification

The framework produces two distinct propagated strengths based on the instantiation of the geometric gate: $S_i^{\text{epi}}$ (from the epistemic stream) and $S_i^{\text{alea}}$ (from the aleatoric stream). We derive the final uncertainty scores as follows:

**Epistemic Uncertainty** ($u^{\text{epi}}$). Epistemic uncertainty is inversely proportional to the accumulated evidence. Using the strength from the epistemic stream, we define:

$$u_i^{\text{epi}} = \frac{K}{S_i^{\text{epi}}}. \qquad (8)$$

Intuitively, if a node is far from the reference set (low $e^{\text{epi}}$), the geometric gate $\lambda$ closes, reducing the total evidence $S_i^{\text{epi}}$,

which directly results in a high epistemic uncertainty score (indicating, for example, an OOD sample).

**Aleatoric Uncertainty ($u^{\mathbf{alea}}$).** Aleatoric uncertainty corresponds to the flatness of the categorical distribution. We use the strength from the aleatoric stream, $S_i^{\text{alea}}$, to construct a rectified posterior probability. First, we isolate informative evidence mass by removing the prior: $e_i^{\text{total}} = S_i^{\text{alea}} - K$. We then define a stream-specific mixing coefficient $v_i^{\text{alea}} = K/S_i^{\text{alea}}$ and compute the expected posterior:

$$\bar{\mathbf{p}}_i^{\text{new}} = \frac{e_i^{\text{total}}\hat{\mathbf{p}}_i + \mathbf{1}}{S_i^{\text{alea}}} = (1 - v_i^{\text{alea}})\hat{\mathbf{p}}_i + \frac{v_i^{\text{alea}}}{K}\mathbf{1}. \quad (9)$$

Here, if a node is close to error anchors (high confusion), $\lambda$ decreases, causing $S_i^{\text{alea}}$ to drop. This increases $v_i^{\text{alea}}$, forcing the distribution $\bar{\mathbf{p}}_i^{\text{new}}$ towards uniformity (maximum entropy). The final score is:

$$u_i^{\text{alea}} = -\max_c \bar{p}_{ic}^{\text{new}}. \quad (10)$$

## 4. Theoretical Analysis

In this section, we provide theoretical motivations and consistency analyses for X-EviProbe, based on latent-space geometry and evidential modeling. Specifically, we analyze: (1) why the $L_1$ nearest-neighbor distance to the reference set can serve as a monotone surrogate controlling support-induced excess risk; (2) why misclassified reference samples can act as sparse anchors for regions with high *intrinsic ambiguity* (aleatoric uncertainty); (3) why the proposed *Holographic Evidence Fusion* provides a coherent hierarchical integration of the three evidence sources; (4) the computational complexity of the framework.

### 4.1. Epistemic Uncertainty: A Lipschitz Excess-Risk Bound

We first justify defining epistemic evidence based on the distance to the *Reference Set* $\mathcal{D}_{\text{ref}}$, interpreting such distance as a measure of missing support in the frozen latent space. Following Assumption 2.1, the frozen classifier is $f = \varphi \circ l \circ g$, where $g$ takes the whole graph $(\mathbf{A}, \mathbf{X})$ and outputs node embeddings $\mathbf{H} = g(\mathbf{A}, \mathbf{X})$; the embedding of node $i$ is the $i$-th row $\mathbf{h}_i$. Let $Y$ denote the (random) node label and let $\mathcal{L}(\cdot, \cdot)$ be the loss function (e.g., cross-entropy). If we view $(H, Y)$ as a generic latent–label pair induced by sampling a node from the data distribution, we define the conditional risk at latent position $\mathbf{h}$ as

$$\mathcal{R}(\mathbf{h}) = \mathbb{E}_{Y|H=\mathbf{h}}\Big[\mathcal{L}\big(\varphi(l(\mathbf{h})), Y\big)\Big]. \quad (11)$$

Although $\mathcal{R}(\mathbf{h})$ is a conditional predictive risk and may include irreducible ambiguity, our epistemic score is tied to the support-induced excess over the reference set risk baseline, rather than to the absolute risk itself.

**Assumption 4.1** ($L_1$-**Lipschitz Continuity in the Latent Space**). There exists a constant $K > 0$ such that the risk function $\mathcal{R} : \mathcal{H} \to \mathbb{R}$ is $K$-Lipschitz continuous with respect to the $L_1$ metric on $\mathcal{H}$:

$$\big|\mathcal{R}(\mathbf{h}_i) - \mathcal{R}(\mathbf{h}_j)\big| \leq K\|\mathbf{h}_i - \mathbf{h}_j\|_1, \quad \forall \mathbf{h}_i, \mathbf{h}_j \in \mathcal{H}. \quad (12)$$

*Theoretical Motivation for $L_1$:* In high-dimensional spaces, the $L_2$ norm suffers from the concentration of measure, diminishing the contrast between nearest and farthest neighbors (Aggarwal et al., 2001). While $L_2$ captures local smoothness for in-distribution data, the $L_1$ norm retains better discriminative power for separating high-risk OOD regions. We therefore adopt $L_1$ universally to ensure robust global detection and a unified metric space.

**Assumption 4.2** (**Uniform Reference Risk Bound After Convergence**). After training converges, the reference set has uniformly bounded risk: there exists a finite constant $\epsilon_{\text{ref}} \geq 0$ such that

$$\sup_{\mathbf{h}_j \in \mathcal{D}_{\text{ref}}} \mathcal{R}(\mathbf{h}_j) \leq \epsilon_{\text{ref}}. \quad (13)$$

**Proposition 4.3** (**Nearest-Neighbor Lipschitz Control**). *Under Assumption 4.1, for any query latent $\mathbf{h}_i$ and its nearest neighbor $\mathbf{h}_{nn} = \arg\min_{\mathbf{h}_j \in \mathcal{D}_{\text{ref}}} \|\mathbf{h}_i - \mathbf{h}_j\|_1$, we have*

$$\mathcal{R}(\mathbf{h}_i) \leq \mathcal{R}(\mathbf{h}_{nn}) + K \|\mathbf{h}_i - \mathbf{h}_{nn}\|_1. \quad (14)$$

*Moreover, under Assumption 4.2, we obtain*

$$\mathcal{R}(\mathbf{h}_i) \leq \epsilon_{ref} + K \|\mathbf{h}_i - \mathbf{h}_{nn}\|_1. \quad (15)$$

*Equivalently, the support-induced excess risk above the reference baseline is bounded by*

$$\big[\mathcal{R}(\mathbf{h}_i) - \epsilon_{ref}\big]_+ \leq K \|\mathbf{h}_i - \mathbf{h}_{nn}\|_1. \quad (16)$$

**Intuition.** The Lipschitz property implies each reference node induces an explicit upper bound on the query risk of the form *(reference risk)+(distance)*. Under the uniform reference risk bound, $\epsilon_{\text{ref}}$ acts as a reference-set baseline, while the distance term controls the support-induced excess risk beyond this baseline. Therefore, the nearest-neighbor distance provides a monotone proxy for epistemic uncertainty caused by missing latent support (Appendix D.1).

### 4.2. Aleatoric Uncertainty via Boundary Sampling

Aleatoric uncertainty corresponds to regions of *intrinsic ambiguity* (irreducible error / Bayes risk), where the conditional label distribution is inherently noisy or overlapping. We justify identifying such regions using the set of misclassified reference samples $\mathcal{M}_{\text{ref}}$.

**Assumption 4.4** (**Optimization Trajectory as Implicit Sampling**). Following stochastic optimization dynamics

(e.g., SGD with mini-batch noise), late-stage training explores a neighborhood of near-optimal parameters. Selecting a checkpoint (e.g., by early stopping on a validation split) can be interpreted as picking one representative hypothesis from this implicit ensemble.

**Proposition 4.5** (**Confusion Boundary Approximation**). *Let $\mathcal{M}_{ref} \subset \mathcal{D}_{ref}$ be the set of samples misclassified by the selected model $f^*$. When $\mathcal{M}_{ref}$ is non-empty and residual errors after convergence are predominantly associated with irreducible ambiguity rather than reducible model error, $\mathcal{M}_{ref}$ can be viewed as a sparse empirical approximation of points enriched near regions of high intrinsic ambiguity (e.g., class-overlap regions), and proximity to $\mathcal{M}_{ref}$ provides evidence for elevated aleatoric uncertainty.*

**Intuition.** After convergence, residual errors are expected to be enriched near low-margin / overlap regions, especially when reducible model error has been largely minimized; misclassified reference nodes act as sparse "error anchors," and proximity to them indicates elevated ambiguity-oriented uncertainty (Appendix D.2).

### 4.3. Holographic Evidence Fusion: Coherent Hierarchical Integration

Finally, we analyze the *Holographic Evidence Fusion*: $\boldsymbol{\alpha}_i = \lambda_i E_i \hat{\mathbf{p}}_i + \mathbf{1}$, where $E_i = \sum_c \text{softplus}(z_{ic})$ is the Logit Magnitude. This construction adheres to a data-flow hierarchy: Geometry $\rightarrow$ Magnitude $\rightarrow$ Direction.

**Proposition 4.6** (**Hierarchical Consistency**). *The fusion mechanism integrates latent geometry, logit magnitude, and predictive probability in a manner aligned with the model's inference hierarchy ($g \rightarrow l \rightarrow \varphi$). This hierarchical integration yields a coherent evidential construction: latent geometry acts as a gate for whether the frozen logit map should contribute evidence, while logit magnitude and predictive probability respectively determine the total evidential strength and its class-wise allocation, supporting the separation of epistemic and aleatoric uncertainty streams.*

**Intuition.** The gate $\lambda_i$ returns a uniform prior whenever the corresponding geometric signal vetoes the native evidence ($\lambda_i \rightarrow 0$); otherwise ($\lambda_i \rightarrow 1$), logit magnitude sets the overall evidence strength and probabilities allocate it, yielding a consistent epistemic/aleatoric decomposition (Appendix D.3).

### 4.4. Complexity Analysis

**Summary.** Beyond the single forward pass of the frozen backbone, X-EviProbe mainly incurs (i) nearest-neighbor distance computation in the latent space, $O(N \cdot N_{ref} \cdot d)$, and (ii) scalar evidence-strength propagation over edges, $O(T \cdot |\mathcal{E}|)$. Full details are provided in Appendix D.4.

## 5. Experiments

In this section, we evaluate the effectiveness of X-EviProbe. Our evaluation is structured into two parts: (1) quantitative experiments to verify its UQ performance against state-of-the-art baselines, and (2) ablation studies to analyze the contribution of each design choice in X-EviProbe. Specifically, we aim to answer the following two research questions:

- **RQ1 (Performance)**: Does parameter-free X-EviProbe **outperform** baselines that rely on auxiliary training, heavy sampling, or black-box mechanisms in quantifying both aleatoric and epistemic uncertainties?

- **RQ2 (Ablation)**: How do different design choices (e.g., neighbor aggregation strategy, kernel type, reference set construction, and key components) affect uncertainty estimation performance?

### 5.1. Experimental Setup

**Datasets.** We conduct comprehensive experiments on seven widely used graph datasets: three citation networks (CoraML, CiteSeer, PubMed), two co-purchase networks (AmazonComputers, AmazonPhoto), and two co-author networks (CoauthorCS, CoauthorPhysics).

**Baselines.** We compare X-EviProbe with 9 competitive baseline models: (1) **Deterministic**: VGNN-entropy, VGNN-max-score; (2) **Sampling-based**: VGNN-dropout (Gal & Ghahramani, 2016), VGNN-ensemble (Lakshminarayanan et al., 2017); (3) **Energy-based**: VGNN-energy (Liu et al., 2020a), VGNN-gnnsafe (Wu et al., 2023); (4) **Evidential**: SGNN-GKDE (Zhao et al., 2020), EGNN (Sensoy et al., 2018), and EPN (Yu et al., 2025).

**Evaluation Protocols.** Aligned with established benchmarks (Zhao et al., 2020; Yu et al., 2025), we assess the quality of uncertainty estimates through two downstream tasks: (1) **OOD Detection**: We employ *epistemic uncertainty* to identify samples that are distributionally shifted from the training manifold. We adopt the "Left-Out-Classes" (LOC) protocol with five different split settings (OS-1 to OS-5) to simulate diverse shifts. For instance, OS-1 and OS-2 designate the last and first classes as OOD, respectively, while OS-3 to OS-5 involve random class subsets. Detailed configurations are provided in Appendix B. (2) **Misclassification Detection**: We leverage *aleatoric uncertainty* to detect potential failures on in-distribution data that arise from inherent data ambiguity, such as class overlap or measurement noise. We primarily report Area Under the Receiver Operating Characteristic curve (AUROC) for both tasks, with Area Under the Precision-Recall Curve (AUPRC) results provided in Appendix F.

**Implementation Details.** For all methods, we employ a

*Table 2.* OOD Detection: AUROC (↑) averaged over five Left-Out-Classes (LOC) settings for each method across 7 datasets. The best performance is marked in **bold**, while the runner-up is underlined. The last row illustrates the percentage improvement of our proposed X-EviProbe compared to the best-performing baseline. Detailed per-setting results are provided in Appendix F.

| Model | CoraML | CiteSeer | PubMed | Amazon Photo | Amazon Computers | Coauthor CS | Coauthor Physics |
|---|---|---|---|---|---|---|---|
| VGNN-entropy | 88.29±2.54 | 87.12±1.69 | 58.45±5.14 | 81.55±4.85 | 72.70±4.54 | 90.48±2.28 | 89.59±3.29 |
| VGNN-max-score | 87.50±2.43 | 86.57±1.76 | 58.45±5.14 | 81.49±4.93 | 71.99±3.38 | 90.13±2.28 | 89.15±3.27 |
| VGNN-dropout | 87.38±2.75 | 85.76±2.86 | 61.85±2.86 | 80.91±5.05 | 71.16±3.88 | 89.79±2.43 | 89.40±3.17 |
| VGNN-ensemble | 87.30±3.20 | 86.63±2.39 | 60.24±3.90 | 81.30±3.55 | 69.13±6.00 | 90.38±2.21 | 89.57±3.40 |
| VGNN-energy | 88.65±2.36 | 87.43±1.68 | 58.34±5.22 | 79.36±4.46 | 72.82±5.55 | 89.00±2.52 | 90.26±3.60 |
| VGNN-gnnsafe | 90.77±1.47 | 89.64±2.50 | 58.98±6.49 | 85.67±6.62 | 78.70±6.58 | 93.43±2.41 | 94.22±2.23 |
| SGNN-GKDE | 88.17±2.34 | 86.39±2.38 | 62.43±4.70 | 83.78±4.36 | 71.80±8.43 | 85.24±1.32 | 90.18±1.51 |
| EGNN | 83.40±3.17 | 86.88±1.15 | 62.28±5.84 | 70.13±3.57 | 63.50±7.24 | 86.30±2.40 | 86.92±2.03 |
| EPN | 90.86±2.18 | 89.56±1.78 | 60.43±8.19 | 83.83±4.88 | 66.72±14.48 | 79.90±11.03 | 93.83±2.95 |
| **X-EviProbe** | **94.52±1.96** | **98.19±0.29** | **83.30±5.36** | **97.08±0.42** | **93.35±1.76** | **98.91±0.24** | **97.99±0.91** |
| *Improv. vs Best* | 4.03% | 9.54% | 33.43% | 13.32% | 18.61% | 5.87% | 4.00% |

*Table 3.* Misclassification Detection: AUROC (↑) performance. The best performance is marked in **bold**, while the runner-up is underlined. The last row illustrates the percentage improvement of our proposed X-EviProbe compared to the best-performing baseline. * indicates statistical significance with $p < 10^{-3}$ under a paired two-sided t-test.

| Model | CoraML | CiteSeer | PubMed | Amazon Photo | Amazon Computers | Coauthor CS | Coauthor Physics |
|---|---|---|---|---|---|---|---|
| VGNN-entropy | 82.22±1.29 | 83.47±1.50 | 74.17±2.10 | 84.39±1.71 | 70.81±2.27 | 84.12±0.97 | 87.40±2.00 |
| VGNN-max-score | 83.75±1.20 | 84.67±1.39 | 75.15±1.38 | 86.01±1.59 | 74.63±1.72 | 87.26±0.78 | 88.52±1.83 |
| VGNN-dropout | 81.75±1.22 | 84.78±0.96 | 72.62±3.98 | 86.60±1.09 | 76.05±2.22 | 86.97±0.82 | 90.22±1.05 |
| VGNN-ensemble | 82.94±2.33 | 85.40±0.71 | 72.87±3.43 | 85.46±0.80 | 77.01±2.88 | 87.31±0.70 | 89.08±0.90 |
| VGNN-energy | 80.31±1.54 | 81.66±1.59 | 69.86±3.10 | 76.48±2.47 | 64.87±2.37 | 74.53±1.61 | 83.63±2.39 |
| VGNN-gnnsafe | 80.97±1.47 | 84.87±1.14 | 70.18±3.03 | 74.54±3.46 | 72.12±4.13 | 72.90±1.77 | 84.66±1.69 |
| SGNN-GKDE | 82.17±2.18 | 84.12±1.44 | 73.28±1.28 | 79.10±2.73 | 73.38±5.05 | 84.67±9.43 | 88.64±1.17 |
| EGNN | 84.50±1.88 | 84.38±2.64 | 74.44±0.93 | 85.55±0.95 | 74.37±1.17 | 86.83±0.69 | 88.34±1.12 |
| EPN | 83.71±0.65 | 84.47±0.76 | 76.04±1.11 | 86.07±0.69 | 77.12±2.23 | 87.79±0.77 | 90.33±1.02 |
| **X-EviProbe** | **87.65±0.73*** | **92.07±0.54*** | **79.33±1.21*** | **88.67±1.08*** | **83.86±1.22*** | **88.18±0.65*** | **92.53±0.63*** |
| *Improv. vs Best* | +3.73% | +7.81% | +4.33% | +2.39% | +8.74% | +0.44% | +2.44% |

two-layer GCN as the backbone and adopt recommended hyperparameters from their respective literature. For EPN (Yu et al., 2025), a key evidential post-hoc baseline, we use its official implementation with fine-grained tuning. We observe that tuning EPN on a single OOD split implies that hyperparameters are optimized using the in-distribution classes of that specific setting. When evaluated on other splits, these "known" classes may become "unknown" OOD targets, meaning their distributional properties have already influenced the hyperparameter selection. In contrast, X-EviProbe is parameter-free in the sense of introducing no trainable UQ-specific parameters and requiring no tuning on OOD or misclassification labels, ensuring strict independence from such distribution-specific tuning.

For statistical robustness, all experiments use 5 random model initializations and 5 data splits. For OOD detection, we average results over these runs within each LOC setting (OS-1 to OS-5) and report the mean across the five settings. For misclassification detection, we directly average over the runs. See Appendix C for comprehensive details.

### 5.2. Performance Evaluation (RQ1)

**OOD Detection.** Table 2 reports the AUROC results of OOD detection on seven datasets, where each entry is averaged over five LOC split settings (OS-1 to OS-5). X-EviProbe consistently achieves the best AUROC across all datasets, showing clear advantages over deterministic, sampling-based, energy-based, and evidential baselines.

More importantly, the improvements are not marginal: compared to the strongest baseline on each dataset, X-EviProbe yields relative gains ranging from 4.00% to 33.43%. Consistently, we also observe strong improvements in AUPR, with relative gains ranging from 3.93% to 48.4% (reported in the appendix). These consistent gains directly answer **RQ1**: as a transparent and parameter-free post-hoc approach, X-EviProbe can deliver state-of-the-art epistemic uncertainty for OOD detection without auxiliary training or heavy sampling, by decoding the frozen latent geometry and the model's own outputs into class-wise Dirichlet evidence.

Among all baselines, **VGNN-gnnsafe** achieves the runner-up performance on most datasets. This suggests that *even a*

*Table 4.* Ablation Study (OOD Detection): AUROC (↑) averaged over five LOC settings. We report results across 7 datasets.

| Variant | CoraML | CiteSeer | PubMed | Amazon Photo | Amazon Computers | Coauthor CS | Coauthor Physics |
|---|---|---|---|---|---|---|---|
| **X-EviProbe (full)** | 94.52±1.96 | 98.19±0.29 | 83.30±5.36 | 97.08±0.42 | 93.35±1.76 | 98.91±0.24 | 97.99±0.91 |
| **Neighbor aggregation:** Top-5 mean (vs. nearest) | 92.15±2.73 | 94.91±0.83 | 74.23±4.62 | 96.63±0.45 | 92.57±1.99 | 97.74±0.96 | 96.80±1.16 |
| Top-10 mean (vs. nearest) | 90.37±3.56 | 91.18±1.59 | 70.37±3.45 | 96.31±0.68 | 92.03±2.24 | 96.91±1.39 | 96.14±1.27 |
| **Kernel:** Gaussian (vs. Laplace) | 93.58±2.14 | 96.68±0.41 | 81.09±5.04 | 97.03±0.35 | 93.38±1.71 | 98.65±0.29 | 97.74±1.07 |
| **Reference set:** train-only (vs. train+val) | 84.19±4.71 | 78.28±2.98 | 58.54±4.76 | 91.77±2.50 | 81.70±4.32 | 93.52±1.41 | 90.53±3.49 |
| val-only (vs. train+val) | 94.38±1.98 | 97.99±0.28 | 83.27±5.36 | 97.06±0.41 | 93.36±1.74 | 98.88±0.26 | 97.99±0.91 |
| **w/o geometry** | 90.78±1.80 | 89.71±2.37 | 58.94±6.67 | 81.94±6.09 | 76.32±9.21 | 85.22±5.98 | 94.00±2.35 |
| **w/o logits** | 92.89±2.45 | 96.73±0.33 | 87.36±4.96 | 97.25±0.55 | 93.76±1.65 | 98.99±0.24 | 97.33±2.03 |
| **w/o prob** | 94.52±1.96 | 98.19±0.29 | 83.30±5.36 | 97.08±0.42 | 93.35±1.76 | 98.91±0.24 | 97.99±0.91 |
| **w/o propagation** | 78.87±2.57 | 81.55±1.98 | 60.76±3.89 | 90.55±0.76 | 84.16±1.64 | 91.79±1.89 | 90.28±3.05 |

*relatively simple uncertainty signal combined with graph-level propagation* can already be effective for OOD detection. In contrast, introducing additional black-box components and split-specific hyperparameter tuning can make the detection behavior less predictable. For example, **EPN** shows noticeably higher variance on several datasets (e.g., Amazon Computers and Coauthor CS), indicating that its performance is sensitive to the particular OOD configuration used for tuning. When transferred to other LOC splits, the tuned hyperparameters may no longer align with the new "unknown" classes, leading to unstable AUROC.

**Misclassification Detection.** Table 3 reports the AUROC results of misclassification detection on seven datasets. X-EviProbe consistently achieves the best AUROC across all datasets, demonstrating clear advantages over deterministic, sampling-based, energy-based, and evidential baselines.

X-EviProbe yields relative gains ranging from +0.44% to +8.74%. Consistently, we also observe strong improvements in AUPR, with relative gains ranging from 4.59% to 46.1% (reported in the appendix). Among the baselines, the strongest competitors vary across datasets (e.g., EGNN/EPN or sampling-based variants), indicating that output-based heuristics (entropy/max-score) or post-hoc evidential modeling alone may not robustly separate correct and incorrect predictions. In contrast, X-EviProbe explicitly down-weights evidence, which helps suppress over-confident evidence in ambiguous areas and yields a more calibrated uncertainty signal for misclassification detection.

### 5.3. Ablation Study (RQ2)

To answer **RQ2**, we ablate the key design choices in X-EviProbe. Table 4 summarizes the OOD (epistemic) ablations, while the misclassification (aleatoric) ablations are reported in Appendix E (Table 7). Overall, we observe three consistent trends: (i) replacing the nearest-neighbor evidence with Top-$k$ averaging hurts performance (larger

$k$ worsens), (ii) Gaussian kernels trade slightly better misclassification performance for worse OOD detection, and (iii) removing geometry or propagation causes the largest degradation, confirming that latent geometry and graph propagation are both crucial. Other output-based factors exhibit stream- and task-specific effects.

To complement the main evaluation, Appendix G reports additional baseline comparisons, backbone generalization experiments, results on a large-scale benchmark, stress tests under distorted geometry, and runtime/ablation results.

## 6. Related Work

Our work addresses post-hoc, evidential, and interpretable uncertainty quantification for frozen, pre-trained GNNs.

**Uncertainty Quantification in GNNs.** Existing approaches are broadly divided into sampling-based and deterministic methods. Sampling methods, such as *Bayesian GNNs* (Pal et al., 2019; Zhang et al., 2019) and *Deep Ensembles* (Lakshminarayanan et al., 2017), estimate uncertainty via prediction variance. Recent variants like *DropEdge* (Rong et al., 2020) and *GNSDE* (Lin et al., 2024; Bergna et al., 2025) adapt this to graph topologies. While effective, they incur high computational costs and require access to training optimization. Deterministic methods, notably *Evidential Deep Learning (EDL)* (Sensoy et al., 2018), improve efficiency by modeling class probabilities as Dirichlet distributions. Approaches like *GPN* (Stadler et al., 2021) and other evidential GNNs (Zhao et al., 2020) extend this to graphs using latent density estimation. However, these methods necessitate specialized architectural constraints or loss functions, rendering them unsuitable for frozen, pre-trained models.

**Post-hoc Uncertainty Estimation.** To bypass retraining, post-hoc methods estimate uncertainty from fixed representations. *Temperature Scaling* (Guo et al., 2017) improves calibration but fails to capture epistemic uncertainty on

OOD data. *Latent Density Models* (Sun et al., 2023) utilize kernel density estimation for uncertainty categorization. Most relevantly, *Evidential Uncertainty Probes* (Yu et al., 2025) train auxiliary networks to predict evidence from frozen embeddings. While flexible, this introduces a "black-box within a black-box" issue, obscuring the true source of uncertainty. In contrast, **X-EviProbe** constructs class-wise Dirichlet evidence directly from the frozen model via latent geometry (reference-set distance and proximity to error anchors) and a principled geometry–logits–probability fusion, without auxiliary models or retraining.

**Explainable Uncertainty.** While general GNN explainability is well-studied (e.g., *GNNExplainer* (Ying et al., 2019)), explaining uncertainty remains challenging. Methods like *CLUE* (Antorán et al., 2020) and $\delta$-*CLUE* (Ley et al., 2021) use counterfactual optimization to find uncertainty-reducing inputs, while Shapley-value approaches (Watson et al., 2023) attribute uncertainty to features. In contrast, our framework is inherently traceable: it propagates a scalar evidence strength over the graph and admits a finite-step attribution through the propagation matrix, which can map high epistemic/aleatoric uncertainty back to influential neighbors and their evidence sources.

## 7. Conclusion

We introduced **X-EviProbe**, a parameter-free post-hoc framework that equips a frozen GNN with *class-wise* Dirichlet evidence, enabling a transparent and decomposable view of epistemic versus aleatoric uncertainty in a single forward pass. By probing frozen latent geometry and fusing it with the model's native outputs, X-EviProbe yields strong performance on both OOD detection and misclassification detection across seven benchmarks, without expensive sampling or auxiliary black-box models. A limitation is the reliance on a representative reference set and nearest-neighbor computations; future work includes scalable and adaptive reference construction, and extending the probing principle to other structured modalities.

## Acknowledgements

This work is supported by the National Key Research and Development Program (Grant No. 2023YFC3303800), the Natural Science Foundation of China (No. 62372057), BUPT-China Unicom Joint Innovation Center (2025-STHZ-BJYDDX-010) and was partially supported by BUPT Kunpeng&Ascend Center of Cultivation. Sihong Xie was supported by the National Key R&D Program of China (Grant No.2023YFF0725001), the Department of Science and Technology of Guangdong Province (2023CX10X079), the Guangzhou-HKUST(GZ) Joint Funding Program (Grant No.2023A03J0008), and the Education Bureau Guangzhou Municipality.

## Impact Statement

This work aims to improve uncertainty quantification for graph neural networks by helping practitioners identify unreliable predictions and distinguish different sources of uncertainty. Potential benefits include safer deployment in domains where overconfident errors can be costly, such as scientific discovery, cybersecurity, fraud detection, financial risk assessment and forecasting, and healthcare. Beyond graph learning, the core post-hoc evidential probing principle, when used without the graph propagation component, may also be adapted to more general neural network models with frozen representations and outputs, broadening the potential scope of transparent uncertainty estimation. However, uncertainty estimates should not be interpreted as guarantees of correctness, especially when the reference set is biased, incomplete, or unrepresentative of deployment conditions. We recommend careful validation under domain-specific distribution shifts, along with reporting calibration and failure cases, before applying the method in high-stakes settings.

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

# A. Dataset Details

We evaluate our framework on seven standard node classification datasets. The dataset statistics are summarized in Table 5. For data splitting, we follow the protocol in Zhao et al. (Zhao et al., 2020) (also adopted by (Yu et al., 2025)). Specifically, for all datasets we use 20 labeled nodes per class for training; the remaining nodes are split into 20% for testing and the rest for validation. To reduce randomness, we report results averaged over 5 random model initializations and 5 data splits.

*Table 5.* Summary of Dataset Statistics.

| Dataset | Nodes | Edges | Features | Classes | Domain |
|---|---|---|---|---|---|
| CoraML | 2,995 | 16,316 | 2,879 | 7 | Citation |
| CiteSeer | 4,230 | 10,674 | 602 | 6 | Citation |
| PubMed | 19,717 | 88,648 | 500 | 3 | Citation |
| AmazonPhoto | 7,650 | 238,162 | 745 | 8 | Co-purchase |
| AmazonComputers | 13,752 | 491,722 | 767 | 10 | Co-purchase |
| CoauthorCS | 18,333 | 163,788 | 6,805 | 15 | Co-author |
| CoauthorPhysics | 34,493 | 495,924 | 8,415 | 5 | Co-author |

# B. Out-of-Distribution (OOD) Settings

To simulate distributional shifts, we employ the Left-Out-Classes (LOC) protocol following (Yu et al., 2025). In this setup, a subset of classes is designated as OOD and removed from the training and validation sets. During the testing phase, nodes belonging to these unseen classes are introduced to evaluate the model's ability to detect epistemic uncertainty. We consider five distinct LOC configurations (OS-1 to OS-5) to rigorously test the robustness of our method. OS-1 selects the last classes as OOD, OS-2 selects the first classes, and OS-3 through OS-5 select random subsets of classes. The specific class indices used as OOD for each dataset and setting are detailed in Table 6.

*Table 6.* Configuration of Left-Out-Classes (OOD classes) for each dataset and setting.

| Dataset | OS-1 (Last) | OS-2 (First) | OS-3 (Rand) | OS-4 (Rand) | OS-5 (Rand) |
|---|---|---|---|---|---|
| CoraML | [4, 5, 6] | [0, 1, 2] | [0, 2, 4] | [1, 3, 5] | [3, 4, 5] |
| CiteSeer | [4, 5] | [0, 1] | [1, 2] | [3, 4] | [0, 5] |
| PubMed | [2] | [0] | [1] | - | - |
| AmazonPhoto | [5, 6, 7] | [0, 1, 2] | [3, 4, 5] | [1, 4, 6] | [2, 3, 7] |
| AmazonComputers | [5-9] | [0-4] | [2,3,4,5,7] | [1,2,3,6,7] | [2,4,5,8,9] |
| CoauthorCS | [11-14] | [0-3] | [1,2,9,12] | [3,6,10,13] | [2,3,6,10] |
| CoauthorPhysics | [3, 4] | [0, 1] | [2, 3] | [0, 4] | [1, 2] |

# C. Experimental Details

In this section, we provide additional details regarding the implementation of X-EviProbe and the baseline methods. All experiments were conducted on a machine equipped with an Apple M4 Pro processor and 24GB of RAM. The software environment included Python 3.9.6, PyTorch 2.6.0, and PyTorch Geometric 2.6.1.

## C.1. Architectures and Training

Following Yu et al. (Yu et al., 2025), all GNN-based models in the main seven-benchmark evaluation use a two-layer Graph Convolutional Network (GCN) as the backbone architecture unless otherwise specified. The hidden layer dimension is set to 64 for all seven datasets. We use ReLU activation and a dropout rate of 0.5. Models are trained using the Adam optimizer with early stopping based on validation loss and a patience of 50. All other training details follow Yu et al. (Yu et al., 2025). For X-EviProbe, the hidden representation from the first GCN layer is used as the embedding matrix $\mathbf{H}$, and the output of the second GCN layer is used as the node-level logits $\mathbf{z}_i$. For scalar evidence-strength propagation, we fix the number of propagation steps to the number of message-passing layers in the frozen GNN backbone. We do not introduce any additional MLP or auxiliary readout head.

## C.2. Baseline Hyperparameters

We follow the hyperparameter settings recommended in the original papers for all baselines. In particular, for baselines whose uncertainty scores (e.g., aleatoric/epistemic uncertainty used in our AU/EU evaluation) depend on implementation details, we follow the evaluation pipeline in (Yu et al., 2025) and reuse their released `Evidential-Probing-GNN`

codebase to compute the corresponding uncertainty scores whenever available (including **VGNN-gnnsafe**, **SGNN-GKDE**, **EGNN**, and **EPN**). Here VGNN denotes the vanilla GNN backbone equipped with the corresponding uncertainty score.

- **VGNN-dropout**: We use 10 MC dropout samples during inference with a dropout rate of 0.5.

- **VGNN-ensemble**: We train 10 independent GCN models with different initializations.

- **VGNN-energy**: The temperature parameter for energy-based scores is set to $T = 1.0$.

- **EPN**: We use the official implementation from (Yu et al., 2025) and follow its reported hyperparameter-selection protocol. Specifically, the hyperparameters are selected on one representative OOD setting and then fixed for the remaining LOC settings. Under this protocol, some classes used as in-distribution samples during hyperparameter selection may be treated as OOD classes in other LOC settings. In contrast, X-EviProbe is parameter-free and does not require any UQ-specific hyperparameter tuning.

## D. Additional Theoretical Details

### D.1. Proof of Proposition 4.3

*Proof.* For any $\mathbf{h}_j \in \mathcal{D}_{\mathrm{ref}}$, by Assumption 4.1

$$\mathcal{R}(\mathbf{h}_i) \leq \mathcal{R}(\mathbf{h}_j) + K \|\mathbf{h}_i - \mathbf{h}_j\|_1. \tag{17}$$

Thus, each candidate $\mathbf{h}_j$ induces an explicit Lipschitz upper bound $U_j(\mathbf{h}_i) := \mathcal{R}(\mathbf{h}_j) + K \|\mathbf{h}_i - \mathbf{h}_j\|_1$. If the exact reference risks were known, the tightest single-anchor bound would be obtained by minimizing $U_j(\mathbf{h}_i)$ over $j$.

Under Assumption 4.2, we have $\mathcal{R}(\mathbf{h}_j) \leq \epsilon_{\mathrm{ref}}$ for all $j$, hence

$$U_j(\mathbf{h}_i) \leq \bar{U}_j(\mathbf{h}_i) := \epsilon_{\mathrm{ref}} + K \|\mathbf{h}_i - \mathbf{h}_j\|_1.$$

Minimizing this uniform-cap surrogate bound therefore reduces to minimizing the distance term, and the nearest neighbor $\mathbf{h}_{nn}$ yields

$$\epsilon_{\mathrm{ref}} + K \|\mathbf{h}_i - \mathbf{h}_{nn}\|_1 = \min_{\mathbf{h}_j \in \mathcal{D}_{\mathrm{ref}}} \left( \epsilon_{\mathrm{ref}} + K \|\mathbf{h}_i - \mathbf{h}_j\|_1 \right). \tag{18}$$

Combining this with Eq. (17) gives

$$\mathcal{R}(\mathbf{h}_i) \leq \epsilon_{\mathrm{ref}} + K \|\mathbf{h}_i - \mathbf{h}_{nn}\|_1. \tag{19}$$

Equivalently, with $[x]_+ = \max\{x, 0\}$,

$$\left[ \mathcal{R}(\mathbf{h}_i) - \epsilon_{\mathrm{ref}} \right]_+ \leq K \|\mathbf{h}_i - \mathbf{h}_{nn}\|_1. \tag{20}$$

This explains why using only the nearest neighbor is sufficient for minimizing the uniform-cap Lipschitz surrogate bound under the stated assumptions; in contrast, averaging multiple neighbors does not generally decrease this bound unless additional structure on $\mathcal{R}(\cdot)$ is imposed. $\qquad \square$

### D.2. Proof Sketch for Proposition 4.5

*Proof Sketch.* Under the widely observed *Neural Collapse* phenomenon (Papyan et al., 2020), latent representations of different classes tend to form compact clusters while the classifier boundary separates these clusters. Points that remain misclassified after convergence are therefore expected to be enriched near class-overlap/low-margin regions—i.e., locations where the learned features provide insufficient margin to distinguish labels, often due to overlap or noise, and are thus consistent with irreducible residual errors when reducible model error has been largely minimized.

Assumption 4.4 provides the intuition: among many near-optimal hypotheses visited late in training, points that persistently lie near ambiguous boundaries are more likely to be misclassified by the selected $f^*$. Hence, the observed set $\mathcal{M}_{\mathrm{ref}}$ acts as empirical anchors for such confusion regions.

For a query embedding $\mathbf{h}_i$, if $\min_{j \in \mathcal{M}_{\mathrm{ref}}} \|\mathbf{h}_i - \mathbf{h}_j\|_1$ is small, then by the Lipschitz continuity of the risk function (Assumption 4.1), $\mathcal{R}(\mathbf{h}_i) \geq \mathcal{R}(\mathbf{h}_j) - K \|\mathbf{h}_i - \mathbf{h}_j\|_1$ for a nearby anchor $\mathbf{h}_j \in \mathcal{M}_{\mathrm{ref}}$. Interpreting the empirical misclassification

of $\mathbf{h}_j$ as evidence of elevated local risk, a smaller distance to $\mathcal{M}_{\mathrm{ref}}$ yields a stronger lower-bound surrogate, providing evidence that $\mathbf{h}_i$ lies in a similarly ambiguous region. Under the convergence premise in Assumption 4.4, the reducible error component is expected to be reduced; to the extent that the remaining residual errors are dominated by intrinsic irreducible error, local proximity to $\mathcal{M}_{\mathrm{ref}}$ serves as a theoretically motivated proxy for *aleatoric uncertainty*. $\square$

### D.3. Proof of Proposition in Section 4.3

*Proof.* We analyze the behavior of the total evidence $S_i = \sum_c \alpha_{ic} = \lambda_i E_i + K$ and the resulting belief distribution under limiting conditions. Here, $\lambda_i$ denotes the stream-specific geometric gate, instantiated as $e_i^{\mathrm{epi}}$ for the epistemic stream and $1 - e_i^{\mathrm{alea}}$ for the aleatoric stream.

**1. Geometric Veto:** When $\lambda_i \to 0$, the stream-specific geometric signal indicates that the frozen logit evidence should be suppressed, either because the node lacks reference support in the epistemic stream or because it is close to error anchors in the aleatoric stream.

$$\boldsymbol{\alpha}_i \to 0 \cdot E_i \hat{\mathbf{p}}_i + \mathbf{1} = \mathbf{1}.$$

Regardless of the logit map's output ($E_i$ or $\hat{\mathbf{p}}_i$), the evidence is nullified and the distribution reverts to the uniform prior. *Implication:* If the geometric gate vetoes the native evidence, the classifier's apparent confidence should not be trusted. In the epistemic stream, this yields maximal epistemic uncertainty ($K/S_i \to 1$); in the aleatoric stream, the posterior categorical distribution becomes uniform and its entropy is maximized, reflecting the absence of reliable class-specific evidence.

**2. Geometric Support (Conditional Trust):** When $\lambda_i \to 1$, the stream-specific geometric signal does not veto the native evidence. The gate opens, and uncertainty quantification is delegated to the frozen logit map's output:

$$\boldsymbol{\alpha}_i \approx E_i \hat{\mathbf{p}}_i + \mathbf{1}.$$

This allows fine-grained differentiation based on the *Logit Magnitude $E_i$*, which serves as a commonly used indicator of model confidence or in-distribution support (Liu et al., 2020a):

- **High Epistemic Uncertainty:** If $E_i$ is low, total evidence $S_i$ remains small. This signals that the model has limited accumulated evidence despite the geometric gate being open.

- **High Aleatoric Uncertainty:** If $E_i$ is high but $\hat{\mathbf{p}}_i$ is flat (high entropy), it corresponds to strong but conflicting class evidence. This signals intrinsic ambiguity.

Thus, the three sources are integrated coherently with the inference hierarchy: **Geometry gates, Magnitude quantifies, and Probability allocates evidence.** $\square$

### D.4. Complexity Details

The computational complexity of X-EviProbe is governed by three steps. Let $N$ be the number of nodes in the graph, $N_{\mathrm{ref}} = |\mathcal{D}_{\mathrm{ref}}|$ the size of the reference set, $d$ the embedding dimension, and $|\mathcal{E}|$ the number of edges.

(1) **Base Inference:** Since X-EviProbe is post-hoc, it requires a single forward pass of the frozen GNN backbone to obtain embeddings and logits for all $N$ nodes. This is a sunk cost shared by all methods. (2) **Evidence Extraction:** We compute $L_1$ distances from all $N$ nodes to the reference set. The complexity is dominated by $O(N \cdot N_{\mathrm{ref}} \cdot d)$. Note that distances to error anchors ($\mathcal{M}_{\mathrm{ref}} \subset \mathcal{D}_{\mathrm{ref}}$) are obtained by simple masking, incurring no additional heavy computation. This step is fully parallelizable on GPUs. (3) **Propagation:** We propagate the **scalar** evidence strength $\mathbf{S} \in \mathbb{R}^N$ over the full graph. Each step costs one sparse matrix-vector multiplication, i.e., $O(|\mathcal{E}|)$. Over $T$ steps, the cost is $O(T \cdot |\mathcal{E}|)$. The total additional computational cost introduced by X-EviProbe (excluding base inference) is $O(N \cdot N_{\mathrm{ref}} \cdot d + T|\mathcal{E}|)$.

## E. Ablation Study for Misclassification Detection

We provide the full ablation results for misclassification detection (aleatoric uncertainty) in Table 7.

*Table 7.* Ablation Study (Misclassification Detection): AUROC (↑). We report results across 7 datasets.

| Variant | CoraML | CiteSeer | PubMed | Amazon Photo | Amazon Computers | Coauthor CS | Coauthor Physics |
|---|---|---|---|---|---|---|---|
| **X-EviProbe (full)** | 87.65±0.73 | 92.07±0.54 | 79.33±1.21 | 88.67±1.08 | 83.86±1.22 | 88.18±0.65 | 92.53±0.63 |
| **Neighbor aggregation:** Top-5 mean (vs. nearest) | 85.54±0.83 | 87.16±1.05 | 76.19±1.76 | 86.73±1.75 | 79.84±1.87 | 86.60±0.86 | 90.09±1.25 |
| Top-10 mean (vs. nearest) | 85.16±0.88 | 86.28±1.14 | 75.77±1.82 | 86.25±1.85 | 79.07±1.98 | 86.17±0.90 | 89.56±1.42 |
| **Kernel:** Gaussian (vs. Laplace) | 87.93±0.79 | 92.08±0.51 | 79.42±1.16 | 89.12±0.91 | 84.92±1.05 | 88.36±0.67 | 92.74±0.60 |
| **Reference set:** train-only (vs. train+val) | 84.80±1.03 | 85.50±1.21 | 75.29±1.90 | 85.67±1.97 | 78.08±2.33 | 85.60±1.14 | 88.85±1.60 |
| val-only (vs. train+val) | 87.62±0.67 | 91.90±0.55 | 79.33±1.21 | 88.66±1.08 | 83.78±1.27 | 88.15±0.64 | 92.52±0.63 |
| **w/o geometry** | 84.60±0.97 | 85.37±1.28 | 75.25±1.88 | 85.34±2.18 | 77.24±2.22 | 85.13±0.97 | 88.53±1.82 |
| **w/o logits** | 87.66±0.77 | 92.74±0.37 | 79.99±0.82 | 90.07±0.81 | 84.48±0.99 | 89.54±0.54 | 93.41±0.48 |
| **w/o prob** | 86.07±1.07 | 91.01±0.74 | 76.52±1.90 | 82.46±2.21 | 77.66±1.70 | 79.88±1.04 | 89.35±1.37 |
| **w/o propagation** | 84.39±1.03 | 90.09±0.58 | 74.02±1.74 | 84.88±1.29 | 74.48±1.42 | 85.26±0.94 | 88.92±1.42 |

# F. Detailed Experimental Results

We provide the detailed quantitative results for OOD detection and Misclassification detection below.

**OOD Detection Results.** Tables 8–12 report the OOD detection scores (AUROC / AUPR) under each Left-Out-Classes setting (OS-1 to OS-5).

*Table 8.* OOD Detection: AUPR (↑) averaged over five Left-Out-Classes (LOC) settings for each method across 7 datasets. The best performance is marked in **bold**, while the runner-up is underlined. The last row illustrates the percentage improvement of our proposed X-EviProbe compared to the best-performing baseline.

| Model | CoraML | CiteSeer | PubMed | Amazon Photo | Amazon Computers | Coauthor CS | Coauthor Physics |
|---|---|---|---|---|---|---|---|
| VGNN-entropy | 82.39±8.44 | 73.97±3.51 | 39.32±12.75 | 70.61±7.64 | 72.21±15.85 | 76.04±10.84 | 80.73±10.78 |
| VGNN-max-score | 81.11±8.42 | 72.48±3.69 | 39.32±12.75 | 70.80±7.05 | 71.76±16.16 | 75.52±10.24 | 78.88±11.66 |
| VGNN-dropout | 80.71±8.70 | 71.78±4.58 | 41.93±11.45 | 69.70±8.99 | 71.35±15.52 | 75.04±10.51 | 79.53±10.32 |
| VGNN-ensemble | 80.57±9.68 | 72.64±4.70 | 40.52±12.75 | 70.33±8.78 | 69.02±17.94 | 75.80±10.41 | 79.95±11.20 |
| VGNN-energy | 83.11±7.74 | 73.84±4.33 | 39.30±12.98 | 68.55±8.49 | 72.16±15.98 | 74.32±12.46 | 83.16±9.60 |
| VGNN-gnnsafe | 85.83±5.80 | 76.21±5.03 | 39.61±12.17 | 75.33±8.00 | 74.80±13.95 | 83.02±9.25 | 89.15±6.14 |
| SGNN-GKDE | 81.33±8.35 | 69.60±5.51 | 43.25±11.58 | 75.95±8.37 | 70.59±18.12 | 61.06±8.53 | 82.18±8.61 |
| EGNN | 77.22±10.25 | 74.42±3.58 | 45.43±10.81 | 55.20±14.98 | 62.58±21.31 | 70.26±11.45 | 78.41±9.41 |
| EPN | 85.00±7.20 | 75.08±4.32 | 41.27±12.10 | 77.28±8.41 | 69.60±13.44 | 59.00±26.05 | 88.47±4.01 |
| **X-EviProbe** | **89.20±8.23** | **94.27±0.96** | **67.42±1.84** | **89.40±4.58** | **87.70±8.68** | **96.45±1.68** | **96.18±1.41** |
| *Improv. vs Best* | 3.93% | 23.70% | 48.40% | 15.68% | 17.25% | 16.18% | 7.89% |

| Dataset | Method | OS-1 (last) | | OS-2 (first) | | OS-3 (random) | |
|---|---|---|---|---|---|---|---|
| | | OOD-AUROC↑ | OOD-AUPR↑ | OOD-AUROC↑ | OOD-AUPR↑ | OOD-AUROC↑ | OOD-AUPR↑ |
| **CoraML** | *logit based* | | | | | | |
| | VGNN-entropy | 89.87±0.99 | 85.50±1.68 | 87.30±1.07 | 79.03±2.06 | 90.31±1.38 | 90.51±1.55 |
| | VGNN-max-score | 89.21±1.04 | 84.23±1.53 | 85.95±1.10 | 77.21±2.11 | 89.35±1.76 | 89.21±2.22 |
| | VGNN-energy | 90.13±1.05 | 85.93±2.02 | 88.47±0.92 | 81.03±1.75 | 90.47±1.71 | 90.87±1.92 |
| | VGNN-gnnsafe | 91.26±0.93 | 86.69±2.12 | 89.27±2.26 | 82.37±4.11 | 90.25±2.41 | 90.51±2.62 |
| | VGNN-dropout | 88.70±1.03 | 83.97±1.99 | 86.39±0.85 | 76.89±2.06 | 89.38±0.61 | 88.91±0.76 |
| | VGNN-ensemble | 89.19±0.85 | 84.51±1.94 | 85.62±1.70 | 75.10±2.81 | 89.34±1.02 | 88.63±1.53 |
| | *evidential based* | | | | | | |
| | SGNN-GKDE | 89.23±1.40 | 84.19±2.13 | 87.06±3.07 | 76.39±5.85 | 89.72±1.70 | 89.21±2.28 |
| | EGNN | 85.14±1.68 | 81.47±2.00 | 83.30±2.89 | 74.19±3.11 | 86.22±1.63 | 87.35±1.72 |
| | EPN | 92.46±0.89 | 88.48±1.69 | 88.16±1.62 | 78.02±3.60 | 90.83±1.95 | 90.00±2.00 |
| | *ours* | | | | | | |
| | **X-EviProbe** | **96.10±0.29** | **93.86±0.64** | **93.01±0.54** | **84.85±1.73** | **95.71±0.25** | **95.68±0.39** |
| **CiteSeer** | *logit based* | | | | | | |
| | VGNN-entropy | 85.92±2.23 | 71.30±2.92 | 90.00±0.58 | 79.11±1.91 | 87.26±2.25 | 76.18±4.13 |
| | VGNN-max-score | 85.18±2.36 | 69.29±4.04 | 89.50±0.62 | 77.69±1.82 | 86.90±2.36 | 75.03±4.62 |
| | VGNN-energy | 86.54±2.31 | 71.36±3.60 | 90.30±0.64 | 80.29±2.06 | 87.50±2.07 | 75.49±3.64 |
| | VGNN-gnnsafe | 89.98±1.98 | 75.11±4.01 | 93.18±0.51 | 83.85±1.74 | 89.07±2.73 | 76.30±5.23 |
| | VGNN-dropout | 84.45±1.11 | 67.04±2.59 | 89.24±0.90 | 77.52±2.38 | 81.90±3.12 | 68.22±3.66 |
| | VGNN-ensemble | 85.87±1.58 | 69.85±3.65 | 89.63±0.48 | 78.24±1.20 | 86.72±1.82 | 74.36±3.89 |
| | *evidential based* | | | | | | |
| | SGNN-GKDE | 86.51±1.20 | 66.07±3.30 | 89.37±1.63 | 76.91±2.71 | 87.23±1.92 | 73.71±3.89 |
| | EGNN | 85.42±2.09 | 69.20±5.30 | 87.98±1.67 | 76.77±3.11 | 85.90±1.26 | 74.71±1.89 |
| | EPN | 90.28±2.05 | 72.49±5.77 | 91.87±0.49 | 81.83±1.63 | 87.56±1.22 | 73.14±2.27 |
| | *ours* | | | | | | |
| | **X-EviProbe** | **98.52±0.36** | **95.07±1.42** | **98.36±0.19** | **94.83±0.87** | **98.22±0.27** | **94.68±1.38** |
| **PubMed** | *logit based* | | | | | | |
| | VGNN-entropy | 63.08±2.32 | 50.81±2.08 | 59.33±6.40 | 25.60±4.31 | 52.93±6.80 | 41.55±3.92 |
| | VGNN-max-score | 63.08±2.32 | 50.82±2.08 | 59.33±6.40 | 25.60±4.31 | 52.93±6.80 | 41.55±3.92 |
| | VGNN-energy | 63.19±2.26 | 51.15±2.12 | 59.04±6.49 | 25.43±4.99 | 52.81±6.91 | 41.32±4.04 |
| | VGNN-gnnsafe | 64.18±2.64 | 51.45±2.65 | 61.05±8.05 | 27.14±7.24 | 51.71±9.68 | 40.25±5.31 |
| | VGNN-dropout | 63.46±1.48 | 51.42±1.67 | 63.55±4.51 | 29.21±3.61 | 58.55±4.91 | 45.17±3.86 |
| | VGNN-ensemble | 63.79±2.07 | 51.85±2.25 | 60.85±3.60 | 26.71±2.93 | 56.07±2.84 | 43.00±1.84 |
| | *evidential based* | | | | | | |
| | SGNN-GKDE | 65.31±1.61 | 53.82±2.81 | 64.97±4.05 | 30.87±3.23 | 57.00±8.45 | 45.04±8.23 |
| | EGNN | 64.21±1.90 | 57.21±2.67 | 66.92±3.13 | 35.96±3.55 | 55.73±3.86 | 43.12±3.73 |
| | EPN | 67.35±2.72 | 54.07±2.65 | 62.54±9.47 | 30.03±8.99 | 51.39±8.47 | 39.72±4.96 |
| | *ours* | | | | | | |
| | **X-EviProbe** | **78.63±2.54** | **67.55±2.60** | **89.15±1.79** | **65.51±4.41** | **82.13±5.00** | **69.19±4.76** |

*Table 9.* OOD detection results (AUROC / AUPR) with GCN as backbone on citation networks for OS-1, OS-2 and OS-3. Bold indicates the best performance, and underline indicates the second best.

| Dataset | Method | OS-1 (last) | | OS-2 (first) | | OS-3 (random) | |
|---|---|---|---|---|---|---|---|
| | | OOD-AUROC↑ | OOD-AUPR↑ | OOD-AUROC↑ | OOD-AUPR↑ | OOD-AUROC↑ | OOD-AUPR↑ |
| | | logit based | | | | | |
| **AmazonPhoto** | VGNN-entropy | 76.29±2.61 | 64.01±3.77 | 78.93±3.61 | 66.48±4.02 | 86.28±2.28 | 70.31±3.65 |
| | VGNN-max-score | 76.26±2.89 | 63.76±3.53 | 79.63±3.96 | 67.62±3.60 | 86.26±2.51 | 71.80±3.88 |
| | VGNN-energy | 76.15±2.73 | 65.17±4.36 | 73.99±4.40 | 63.68±5.70 | 82.04±3.43 | 64.20±5.89 |
| | VGNN-gnnsafe | 77.53±5.03 | 62.03±5.90 | 85.66±6.48 | 78.77±5.75 | 91.85±1.71 | 78.16±3.26 |
| | VGNN-dropout | 73.66±1.91 | 61.46±3.30 | 78.47±4.26 | 65.27±2.68 | 85.85±1.94 | 69.89±3.12 |
| | VGNN-ensemble | 76.06±3.57 | 64.38±4.89 | 79.69±3.40 | 66.75±2.64 | 83.29±2.61 | 67.41±2.89 |
| | | evidential based | | | | | |
| | SGNN-GKDE | 79.70±4.99 | 68.48±7.59 | 78.42±3.37 | 71.04±5.47 | 86.36±3.59 | 73.46±6.92 |
| | EGNN | 70.62±2.39 | 60.64±3.21 | 64.93±2.39 | 47.52±2.59 | 68.54±1.95 | 47.28±1.64 |
| | EPN | 87.79±3.81 | 81.95±4.97 | 79.01±5.58 | 73.63±6.14 | 80.78±8.63 | 64.46±13.89 |
| | | ours | | | | | |
| | **X-EviProbe** | **97.44±0.16** | **90.32±0.25** | **97.57±0.29** | **91.19±0.42** | **96.85±0.30** | **87.80±0.64** |
| | | logit based | | | | | |
| **AmazonComputers** | VGNN-entropy | 71.08±4.01 | 51.09±7.21 | 80.00±2.20 | 90.87±1.34 | 68.23±6.67 | 72.82±5.65 |
| | VGNN-max-score | 69.97±4.15 | 49.55±6.84 | 76.63±2.67 | 89.06±1.67 | 68.91±7.51 | 74.46±7.07 |
| | VGNN-energy | 70.74±4.15 | 50.79±7.90 | 82.61±3.10 | 91.57±1.87 | 68.86±5.30 | 73.78±4.45 |
| | VGNN-gnnsafe | 80.35±2.26 | 58.50±4.37 | 88.05±3.24 | 92.05±1.67 | 69.79±9.31 | 74.47±6.20 |
| | VGNN-dropout | 71.36±5.05 | 52.49±9.92 | 75.74±1.59 | 88.80±1.22 | 66.50±5.59 | 72.27±4.37 |
| | VGNN-ensemble | 66.42±6.43 | 45.92±9.36 | 74.73±2.59 | 88.06±2.01 | 62.46±2.85 | 68.44±2.29 |
| | | evidential based | | | | | |
| | SGNN-GKDE | 68.60±4.55 | 47.26±8.38 | 86.65±2.69 | 93.75±1.56 | 66.01±6.36 | 70.94±6.07 |
| | EGNN | 59.75±1.79 | 35.58±1.56 | 75.55±1.33 | 87.21±0.92 | 59.06±2.86 | 66.52±2.20 |
| | EPN | 78.26±4.50 | 63.06±6.98 | 68.44±9.49 | 83.40±8.08 | 41.62±3.91 | 53.76±2.84 |
| | | ours | | | | | |
| | **X-EviProbe** | **95.31±0.14** | **78.42±0.39** | **94.57±0.81** | **95.37±0.39** | **93.84±1.57** | **92.85±1.15** |
| | | logit based | | | | | |
| **CoauthorCS** | VGNN-entropy | 87.71±1.74 | 83.74±1.71 | 89.86±0.93 | 66.62±3.17 | 89.22±2.50 | 62.95±6.43 |
| | VGNN-max-score | 87.05±1.75 | 83.01±1.74 | 89.94±0.85 | 67.31±3.14 | 88.94±2.26 | 62.33±5.86 |
| | VGNN-energy | 88.02±1.83 | 84.38±2.11 | 87.21±1.23 | 63.05±2.85 | 87.31±3.50 | 60.49±7.76 |
| | VGNN-gnnsafe | 90.03±2.22 | 88.93±1.92 | 93.67±0.82 | 76.25±2.96 | 92.51±2.56 | 71.43±7.93 |
| | VGNN-dropout | 86.87±1.98 | 82.97±2.98 | 88.45±2.45 | 64.90±5.95 | 89.02±2.54 | 62.79±6.99 |
| | VGNN-ensemble | 88.14±1.62 | 83.54±1.31 | 89.68±2.15 | 67.32±5.05 | 88.70±2.25 | 62.44±6.60 |
| | | evidential based | | | | | |
| | SGNN-GKDE | 84.97±2.91 | 73.51±4.58 | 83.43±3.99 | 53.74±7.44 | 87.09±3.12 | 55.98±7.93 |
| | EGNN | 84.81±1.66 | 79.97±2.98 | 84.07±2.12 | 59.79±4.29 | 85.93±2.72 | 58.47±5.66 |
| | EPN | 92.94±1.88 | 90.23±2.56 | 73.08±10.76 | 41.85±16.40 | 72.43±8.66 | 34.25±10.24 |
| | | ours | | | | | |
| | **X-EviProbe** | **98.58±0.21** | **97.98±0.22** | **99.04±0.05** | **95.83±0.34** | **98.95±0.05** | **94.36±0.28** |
| | | logit based | | | | | |
| **CoauthorPhysics** | VGNN-entropy | 93.45±0.81 | 77.11±2.67 | 86.17±4.98 | 75.92±6.60 | 91.76±1.61 | 91.02±1.95 |
| | VGNN-max-score | 92.70±0.78 | 72.23±3.12 | 85.68±5.02 | 73.74±6.82 | 92.10±1.61 | 91.77±1.85 |
| | VGNN-energy | 94.42±0.85 | 81.51±2.15 | 86.34±5.18 | 77.14±7.05 | 90.92±2.12 | 90.89±2.46 |
| | VGNN-gnnsafe | 96.41±0.44 | 87.40±1.47 | 91.41±4.06 | 84.38±5.55 | 94.56±1.39 | 94.13±1.91 |
| | VGNN-dropout | 92.46±1.30 | 71.59±4.04 | 84.56±2.80 | 71.10±3.96 | 92.04±1.83 | 91.56±1.92 |
| | VGNN-ensemble | 93.23±0.57 | 74.86±2.22 | 86.66±4.53 | 74.48±6.73 | 92.53±0.59 | 92.20±0.55 |
| | | evidential based | | | | | |
| | SGNN-GKDE | 91.39±1.92 | 73.13±4.52 | 89.21±3.97 | 79.79±6.03 | 89.97±2.70 | 88.38±3.47 |
| | EGNN | 89.77±0.92 | 67.48±2.86 | 86.91±2.74 | 75.42±5.00 | 85.67±2.04 | 87.20±1.80 |
| | EPN | 96.79±0.70 | 87.19±2.54 | 94.45±2.10 | 88.19±3.70 | 88.87±4.30 | 86.66±4.74 |
| | | ours | | | | | |
| | **X-EviProbe** | **98.60±0.17** | **94.33±0.51** | **98.38±0.42** | **95.97±0.79** | **96.50±0.43** | **96.21±0.64** |

*Table 10.* OOD detection results (AUROC / AUPR) with GCN as backbone on co-purchase and co-author networks for OS-1, OS-2 and OS-3. Bold indicates the best performance, and underline indicates the second best.

| Dataset | Method | OS-4 (random) | | OS-5 (random) | |
|---|---|---|---|---|---|
| | | OOD-AUROC↑ | OOD-AUPR↑ | OOD-AUROC↑ | OOD-AUPR↑ |
| **CoraML** | **logit based** | | | | |
| | VGNN-entropy | 84.27±3.21 | 69.32±6.17 | 89.70±1.17 | 87.60±1.54 |
| | VGNN-max-score | 83.96±3.17 | 68.35±6.26 | 89.03±1.32 | 86.58±1.85 |
| | VGNN-energy | 84.66±3.32 | 70.77±6.55 | 89.53±1.20 | 86.95±1.60 |
| | VGNN-gnnsafe | 90.00±1.69 | **77.80±4.90** | 93.07±0.70 | 91.76±1.06 |
| | VGNN-dropout | 82.98±2.81 | 67.37±4.31 | 89.43±0.86 | 86.40±1.43 |
| | VGNN-ensemble | 82.43±1.64 | 66.29±2.42 | 89.92±1.00 | 88.34±1.26 |
| | **evidential based** | | | | |
| | SGNN-GKDE | 84.59±2.95 | 69.27±4.91 | 90.25±1.46 | 87.58±1.93 |
| | EGNN | 78.06±3.42 | 60.91±4.12 | 84.27±2.20 | 82.17±2.61 |
| | EPN | 89.34±1.76 | 76.50±5.09 | 93.48±0.80 | 92.01±1.25 |
| | **ours** | | | | |
| | **X-EviProbe** | **91.85±0.81** | 76.72±2.00 | **95.91±0.52** | **94.88±0.86** |
| **CiteSeer** | **logit based** | | | | |
| | VGNN-entropy | 86.12±1.04 | 71.51±2.47 | 86.29±3.69 | 71.78±6.05 |
| | VGNN-max-score | 85.50±1.13 | 70.55±2.33 | 85.76±3.89 | 69.86±6.48 |
| | VGNN-energy | 86.16±0.82 | 68.94±3.09 | 86.66±3.39 | 73.13±5.33 |
| | VGNN-gnnsafe | 86.16±1.74 | 69.75±5.68 | 89.77±2.58 | 76.04±5.04 |
| | VGNN-dropout | 87.72±1.18 | 75.58±3.20 | 85.51±1.77 | 70.55±4.57 |
| | VGNN-ensemble | 87.76±1.16 | 74.62±2.81 | 83.17±5.41 | 66.14±6.44 |
| | **evidential based** | | | | |
| | SGNN-GKDE | 86.02±1.42 | 67.52±3.32 | 82.80±3.92 | 63.76±5.74 |
| | EGNN | 87.34±1.37 | 78.46±3.07 | 87.75±3.28 | 72.95±4.28 |
| | EPN | 87.98±2.26 | 71.11±6.60 | 90.11±3.21 | 76.83±6.08 |
| | **ours** | | | | |
| | **X-EviProbe** | **97.77±0.13** | **94.10±1.74** | **98.08±0.25** | **92.68±0.83** |

*Table 11.* OOD detection results (AUROC / AUPR) with GCN as backbone on citation networks for OS-4 and OS-5. Bold indicates the best performance, and underline indicates the second best.

| Dataset | Method | OS-4 (random) | | OS-5 (random) | |
|---|---|---|---|---|---|
| | | OOD-AUROC↑ | OOD-AUPR↑ | OOD-AUROC↑ | OOD-AUPR↑ |
| **AmazonPhoto** | *logit based* | | | | |
| | VGNN-entropy | 79.10±3.17 | 83.60±2.57 | 87.14±3.61 | 68.65±2.93 |
| | VGNN-max-score | 78.11±3.22 | 82.33±2.67 | 87.18±3.70 | 68.50±3.16 |
| | VGNN-energy | 79.49±3.20 | 83.64±2.86 | 85.11±3.68 | 66.04±4.71 |
| | VGNN-gnnsafe | 80.78±3.71 | 83.00±3.13 | 92.53±1.66 | 74.69±3.34 |
| | VGNN-dropout | 81.31±3.77 | 84.83±3.16 | 85.28±3.06 | 67.07±3.05 |
| | VGNN-ensemble | 82.24±2.50 | 85.89±1.95 | 85.24±3.25 | 67.23±3.88 |
| | *evidential based* | | | | |
| | SGNN-GKDE | 86.86±2.81 | 89.85±2.44 | 87.58±4.60 | 76.92±7.61 |
| | EGNN | 74.07±2.46 | 78.91±1.99 | 72.49±3.04 | 41.63±3.09 |
| | EPN | 81.35±9.03 | 85.82±5.85 | 90.23±2.91 | 80.55±4.95 |
| | *ours* | | | | |
| | **X-EviProbe** | **96.92±0.13** | **95.03±0.24** | **96.60±0.20** | **82.66±0.42** |
| **AmazonComputers** | *logit based* | | | | |
| | VGNN-entropy | 70.38±5.00 | 62.94±5.11 | 73.80±3.72 | 83.35±2.57 |
| | VGNN-max-score | 69.92±5.86 | 61.86±5.97 | 74.50±4.42 | 83.86±3.42 |
| | VGNN-energy | 70.82±3.76 | 62.71±4.41 | 71.06±2.91 | 81.95±2.01 |
| | VGNN-gnnsafe | 78.64±3.09 | 64.14±3.11 | 76.70±3.35 | 84.86±2.08 |
| | VGNN-dropout | 68.15±3.11 | 59.30±3.58 | 74.06±5.37 | 83.88±3.72 |
| | VGNN-ensemble | 65.84±3.08 | 57.56±3.38 | 76.21±5.47 | 85.12±4.41 |
| | *evidential based* | | | | |
| | SGNN-GKDE | 67.64±3.98 | 59.71±4.94 | 70.11±3.10 | 81.28±2.30 |
| | EGNN | 58.17±2.61 | 46.59±2.30 | 64.97±1.88 | 77.03±1.43 |
| | EPN | 71.50±9.76 | 63.84±7.06 | 73.77±2.73 | 83.92±1.93 |
| | *ours* | | | | |
| | **X-EviProbe** | **91.35±0.44** | **78.07±0.57** | **91.67±0.47** | **93.79±0.30** |
| **CoauthorCS** | *logit based* | | | | |
| | VGNN-entropy | 93.23±1.48 | 87.97±2.37 | 92.39±1.27 | 78.91±2.72 |
| | VGNN-max-score | 92.27±1.51 | 86.07±2.32 | 92.45±1.28 | 78.87±3.20 |
| | VGNN-energy | 93.27±1.46 | 88.48±2.38 | 89.16±1.47 | 75.19±2.35 |
| | VGNN-gnnsafe | 96.58±1.16 | 94.24±1.53 | 94.35±0.96 | 84.22±2.10 |
| | VGNN-dropout | 92.16±0.98 | 85.53±1.54 | 92.46±0.60 | 79.03±1.27 |
| | VGNN-ensemble | 92.15±1.48 | 86.29±2.20 | 93.21±0.37 | 79.41±1.48 |
| | *evidential based* | | | | |
| | SGNN-GKDE | 85.15±4.27 | 66.34±7.55 | 85.57±3.01 | 55.72±6.01 |
| | EGNN | 90.27±1.15 | 83.70±1.89 | 86.40±2.07 | 69.36±3.35 |
| | EPN | 90.87±3.52 | 84.02±6.73 | 70.20±7.54 | 44.64±12.61 |
| | *ours* | | | | |
| | **X-EviProbe** | **99.20±0.12** | **98.38±0.29** | **98.79±0.04** | **95.71±0.25** |
| **CoauthorPhysics** | *logit based* | | | | |
| | VGNN-entropy | 86.22±4.36 | 67.15±10.01 | 90.34±2.34 | 92.47±2.04 |
| | VGNN-max-score | 86.06±4.23 | 65.96±8.98 | 89.21±2.55 | 90.71±2.64 |
| | VGNN-energy | 86.78±5.00 | 71.52±10.09 | 92.84±1.43 | 94.76±1.10 |
| | VGNN-gnnsafe | 92.48±3.59 | 82.89±7.91 | 96.24±0.89 | 96.94±0.79 |
| | VGNN-dropout | 89.19±3.91 | 73.40±7.18 | 88.77±3.08 | 90.01±3.12 |
| | VGNN-ensemble | 85.64±5.39 | 66.97±8.74 | 89.82±3.88 | 91.23±3.85 |
| | *evidential based* | | | | |
| | SGNN-GKDE | 88.34±2.42 | 75.95±4.02 | 91.99±1.82 | 93.66±1.66 |
| | EGNN | 87.78±1.97 | 72.76±3.68 | 84.47±3.75 | 89.20±3.07 |
| | EPN | 94.23±2.16 | 84.97±4.85 | 94.79±2.21 | 95.34±2.00 |
| | *ours* | | | | |
| | **X-EviProbe** | **98.69±0.17** | **96.09±0.53** | **97.80±0.60** | **98.29±0.54** |

*Table 12.* OOD detection results (AUROC / AUPR) with GCN as backbone on co-purchase and co-author networks for OS-4 and OS-5. Bold indicates the best performance, and underline indicates the second best.

**Misclassification Detection Results.** Table 13 shows the AUPR scores for detecting misclassified nodes. Higher scores indicate that the model's aleatoric uncertainty is well-calibrated with its prediction errors.

*Table 13.* Misclassification Detection: AUPR (↑) performance. The best performance is marked in **bold**, while the runner-up is underlined. The last row illustrates the percentage improvement of our proposed X-EviProbe compared to the best-performing baseline. $^*$ indicates statistical significance with $p < 10^{-3}$ under a paired two-sided t-test, and $^\dagger$ indicates the single case with $p < 0.051$. Struck-through entries indicate invalid accuracy.

| Model | CoraML | CiteSeer | PubMed | Amazon Photo | Amazon Computers | Coauthor CS | Coauthor Physics |
|---|---|---|---|---|---|---|---|
| VGNN-entropy | 47.46±2.82 | 51.46±4.23 | 43.09±2.82 | 41.27±4.27 | 33.85±2.60 | 33.58±1.55 | 34.06±2.79 |
| VGNN-max-score | 51.36±3.29 | 54.56±4.42 | 44.47±2.09 | 44.71±4.01 | 39.60±2.97 | 40.92±1.49 | 38.14±2.77 |
| VGNN-dropout | 50.16±2.71 | 52.14±5.04 | 42.63±2.12 | 47.28±3.41 | 41.25±3.52 | 38.74±2.50 | 39.46±2.78 |
| VGNN-ensemble | 52.65±4.30 | 53.99±4.69 | 41.97±1.35 | 44.65±2.72 | 41.57±4.35 | 39.67±1.75 | 37.10±1.06 |
| VGNN-energy | 44.95±3.44 | 49.61±4.68 | 39.96±2.92 | 33.74±4.80 | 30.49±2.31 | 26.89±1.90 | 29.44±2.32 |
| VGNN-gnnsafe | 45.62±3.95 | 55.44±3.96 | 38.72±3.62 | 26.62±5.01 | 40.98±2.96 | 20.54±0.86 | 28.37±1.55 |
| SGNN-GKDE | 50.44±5.60 | 52.23±5.54 | 42.19±1.17 | 46.88±9.06 | ~~68.18±6.05~~ | ~~90.07±6.54~~ | 35.30±2.39 |
| EGNN | 53.08±4.97 | 53.36±6.72 | 42.49±2.66 | 42.75±2.90 | 38.94±3.52 | 39.31±2.06 | 37.43±1.59 |
| EPN | 54.59±4.01 | 52.63±4.06 | 44.21±2.16 | 46.88±5.34 | 40.74±3.22 | 40.96±1.66 | 38.24±1.99 |
| **X-EviProbe** | **61.89±4.01**$^*$ | **81.00±1.51**$^*$ | **48.93±2.19**$^*$ | **49.45±3.89**$^\dagger$ | **51.87±2.39**$^*$ | **43.17±1.69**$^*$ | **47.10±1.58**$^*$ |
| *Improv. vs Best* | +13.37% | +46.10% | +10.03% | +4.59% | +24.78% | +5.40% | +19.36% |

| Dataset | Model | Clean Graph | | |
| --- | --- | --- | --- | --- |
| | | clean-acc↑ | MIS_AUROC↑ | MIS_AUPR↑ |
| CoraML | VGNN-entropy | 81.37±1.21 | 82.22±1.29 | 47.46±2.82 |
| | VGNN-max-score | 81.37±1.21 | 83.75±1.20 | 51.36±3.29 |
| | VGNN-energy | 81.37±1.21 | 80.31±1.54 | 44.95±3.44 |
| | VGNN-gnnsafe | 81.37±1.21 | 80.97±1.47 | 45.62±3.95 |
| | VGNN-dropout | 82.11±1.05 | 81.75±1.22 | 50.16±2.71 |
| | VGNN-ensemble | 81.81±1.79 | 82.94±2.33 | 52.65±4.30 |
| | SGNN-GKDE | 81.41±2.19 | 82.17±2.18 | 50.44±5.60 |
| | EGNN | **82.30±1.75** | 84.50±1.88 | 53.08±4.97 |
| | EPN | 80.59±2.12 | 83.71±0.65 | 54.59±4.01 |
| | **X-EviProbe** | 81.54±1.12 | **87.65±0.73** | **61.89±4.01** |
| CiteSeer | VGNN-entropy | 82.95±1.05 | 83.47±1.50 | 51.46±4.23 |
| | VGNN-max-score | 82.95±1.05 | 84.67±1.39 | 54.56±4.42 |
| | VGNN-energy | 82.95±1.05 | 81.66±1.59 | 49.61±4.68 |
| | VGNN-gnnsafe | 82.95±1.05 | 84.87±1.14 | 55.44±3.96 |
| | VGNN-dropout | 83.81±0.71 | 84.78±0.96 | 52.14±5.04 |
| | VGNN-ensemble | **84.18±0.81** | 85.40±0.71 | 53.99±4.69 |
| | SGNN-GKDE | 83.81±0.86 | 84.12±1.44 | 52.23±5.54 |
| | EGNN | 82.86±1.28 | 84.38±2.64 | 53.36±6.72 |
| | EPN | 83.99±0.85 | 84.47±0.76 | 52.63±4.06 |
| | **X-EviProbe** | 82.95±1.05 | **92.07±0.54** | **81.00±1.51** |
| PubMed | VGNN-entropy | 77.84±0.93 | 74.17±2.10 | 43.09±2.82 |
| | VGNN-max-score | 77.84±0.93 | 75.15±1.38 | 44.47±2.09 |
| | VGNN-energy | 77.84±0.93 | 69.86±3.10 | 39.96±2.92 |
| | VGNN-gnnsafe | 77.84±0.93 | 70.18±3.03 | 38.72±3.62 |
| | VGNN-dropout | 76.47±4.80 | 72.62±3.98 | 42.63±2.12 |
| | VGNN-ensemble | 77.03±3.33 | 72.87±3.43 | 41.97±1.35 |
| | SGNN-GKDE | 78.05±1.40 | 73.28±1.28 | 42.19±1.17 |
| | EGNN | 77.94±2.95 | 74.44±0.93 | 42.49±2.66 |
| | EPN | **78.44±0.95** | 76.04±1.11 | 44.21±2.16 |
| | **X-EviProbe** | 77.84±0.93 | **79.33±1.21** | **48.93±2.19** |
| AmazonPhoto | VGNN-entropy | 90.23±0.55 | 84.39±1.71 | 41.27±4.27 |
| | VGNN-max-score | 90.23±0.55 | 86.01±1.59 | 44.71±4.01 |
| | VGNN-energy | 90.23±0.55 | 76.48±2.47 | 33.74±4.80 |
| | VGNN-gnnsafe | 90.23±0.55 | 74.54±3.46 | 26.62±5.01 |
| | VGNN-dropout | 89.68±1.25 | 86.60±1.09 | 47.28±3.41 |
| | VGNN-ensemble | 90.22±0.77 | 85.46±0.80 | 44.65±2.72 |
| | SGNN-GKDE | 79.99±4.75 | 79.10±2.73 | 46.88±9.06 |
| | EGNN | **90.24±0.98** | 85.55±0.95 | 42.75±2.90 |
| | EPN | 89.24±1.60 | 86.07±0.69 | 46.88±5.34 |
| | **X-EviProbe** | 90.23±0.55 | **88.67±1.08** | **49.45±3.89** |

*Table 14.* Misclassification detection results (clean-acc / MIS_AUROC / MIS_AUPR) on CoraML, CiteSeer, PubMed, and AmazonPhoto. The best performance is marked in **bold**, and the second best is underlined.

| Dataset | Model | Clean Graph | | |
| --- | --- | --- | --- | --- |
| | | clean-acc↑ | MIS_AUROC↑ | MIS_AUPR↑ |
| **AmazonComputers** | VGNN-entropy | 81.55±0.85 | 70.81±2.27 | 33.85±2.60 |
| | VGNN-max-score | 81.55±0.85 | 74.63±1.72 | 39.60±2.97 |
| | VGNN-energy | 81.55±0.85 | 64.87±2.37 | 30.49±2.31 |
| | VGNN-gnnsafe | 81.55±0.85 | 72.12±4.13 | 40.98±2.96 |
| | VGNN-dropout | 82.49±0.97 | 76.05±2.22 | 41.25±3.52 |
| | VGNN-ensemble | **82.65±0.35** | 77.01±2.88 | 41.57±4.35 |
| | SGNN-GKDE | ~~55.58±5.59~~ | ~~73.38±5.05~~ | ~~68.18±6.05~~ |
| | EGNN | 81.45±1.21 | 74.37±1.17 | 38.94±3.52 |
| | EPN | 82.55±1.34 | 77.12±2.23 | 40.74±3.22 |
| | **X-EviProbe** | 81.55±0.85 | **83.86±1.22** | **51.87±2.39** |
| **CoauthorCS** | VGNN-entropy | 91.66±0.42 | 84.12±0.97 | 33.58±1.55 |
| | VGNN-max-score | 91.66±0.42 | 87.26±0.78 | 40.92±1.49 |
| | VGNN-energy | 91.66±0.42 | 74.53±1.61 | 26.89±1.90 |
| | VGNN-gnnsafe | 91.66±0.42 | 72.90±1.77 | 20.54±0.86 |
| | VGNN-dropout | 91.79±0.56 | 86.97±0.82 | 38.74±2.50 |
| | VGNN-ensemble | **91.97±0.29** | 87.31±0.70 | 39.67±1.75 |
| | SGNN-GKDE | ~~34.83±6.71~~ | ~~84.67±9.43~~ | ~~90.07±6.54~~ |
| | EGNN | 91.52±0.47 | 86.83±0.69 | 39.31±2.06 |
| | EPN | 91.74±0.25 | 87.79±0.77 | 40.96±1.66 |
| | **X-EviProbe** | 91.66±0.42 | **88.18±0.65** | **43.17±1.69** |
| **CoauthorPhysics** | VGNN-entropy | 92.62±0.42 | 87.40±2.00 | 34.06±2.79 |
| | VGNN-max-score | 92.62±0.42 | 88.52±1.83 | 38.14±2.77 |
| | VGNN-energy | 92.62±0.42 | 83.63±2.39 | 29.44±2.32 |
| | VGNN-gnnsafe | 92.62±0.42 | 84.66±1.69 | 28.37±1.55 |
| | VGNN-dropout | 93.18±0.47 | 90.22±1.05 | 39.46±2.78 |
| | VGNN-ensemble | 93.40±0.55 | 89.08±0.90 | 37.10±1.06 |
| | SGNN-GKDE | **93.71±0.51** | 88.64±1.17 | 35.30±2.39 |
| | EGNN | 92.87±0.79 | 88.34±1.12 | 37.43±1.59 |
| | EPN | 93.47±0.62 | 90.33±1.02 | 38.24±1.99 |
| | **X-EviProbe** | 92.62±0.42 | **92.53±0.63** | **47.10±1.58** |

*Table 15.* Misclassification detection results (clean-acc / MIS_AUROC / MIS_AUPR) on AmazonComputers, CoauthorCS, and Coauthor-Physics. The best performance is marked in **bold**, and the second best is underlined.

# G. Additional Results

This section provides additional evaluations that complement the main experiments, including additional baseline comparisons, large-scale results, backbone generalization, stress tests under distorted latent geometry, runtime comparisons, and additional ablations. Unless otherwise specified, all additional experiments in this appendix follow the evaluation protocols and implementation details of Yu et al. (Yu et al., 2025). We describe only settings that are specific to X-EviProbe or differ from that protocol.

## G.1. Additional Baselines: GPN and GEBM

Tables 16 and 17 compare X-EviProbe with GPN (Stadler et al., 2021) and GEBM (Fuchsgruber et al., 2024). For these additional baseline comparisons, we use the same data splits, LOC settings, and metrics as in Yu et al. (Yu et al., 2025), and follow the corresponding official implementations for method-specific details. These results provide additional reference points for latent-density-based and energy-based uncertainty estimation methods.

*Table 16.* GPN&GEBM Baseline Comparison (OOD Detection). We report OOD detection AUROC (↑) across 7 datasets. The best performance is marked in **bold**.

| Model | CoraML | CiteSeer | PubMed | Amazon Photo | Amazon Computers | Coauthor CS | Coauthor Physics |
|---|---|---|---|---|---|---|---|
| GPN | 86.03±4.64 | 87.50±3.67 | 61.05±4.90 | 88.70±3.92 | 80.53±5.96 | 85.39±4.70 | 84.47±10.32 |
| GEBM | 90.13±2.54 | 88.71±1.47 | 63.93±3.46 | 83.40±2.42 | 75.81±8.13 | 92.20±1.93 | 91.16±3.85 |
| **X-EviProbe** | **94.52±1.96** | **98.19±0.29** | **83.30±5.36** | **97.08±0.42** | **93.35±1.76** | **98.91±0.24** | **97.99±0.91** |

*Table 17.* GPN&GEBM Baseline Comparison (Misclassification Detection). We report misclassification detection MIS_AUROC (↑) across 7 datasets. The best performance is marked in **bold**.

| Model | CoraML | CiteSeer | PubMed | Amazon Photo | Amazon Computers | Coauthor CS | Coauthor Physics |
|---|---|---|---|---|---|---|---|
| GPN | 84.17±1.90 | 86.58±1.26 | 75.66±1.98 | 82.46±2.32 | 78.98±2.09 | 84.54±1.47 | 89.49±1.02 |
| GEBM | 78.35±3.16 | 83.62±1.74 | 65.10±2.41 | 67.82±3.03 | 65.85±4.73 | 72.92±2.92 | 86.82±1.90 |
| **X-EviProbe** | **87.65±0.73** | **92.07±0.54** | **79.33±1.21** | **88.67±1.08** | **83.86±1.22** | **88.18±0.65** | **92.53±0.63** |

## G.2. Large-scale Evaluation on OGBN-Arxiv

For OGBN-Arxiv, we follow the large-scale evaluation protocol of Yu et al. (Yu et al., 2025), including its public split and OGBN-specific backbone/training settings. Tables 18 and 19 report results on OGBN-Arxiv. We include both the full reference set and a 20K-reference subsampling variant to examine the scalability of nearest-neighbor probing. For misclassification detection, we additionally report the w/o-logits variant, following the ablation observation that logit magnitude can occasionally amplify confidently wrong predictions.

*Table 18.* OGBN-Arxiv Full Results (incl. Subsampling): OOD Detection. We report OOD-AUROC (↑) on `ogbn-arxiv`. The best performance is marked in **bold**. EPN is reported under its split-specific tuning protocol, while X-EviProbe is parameter-free and does not tune on OOD splits.

| Model | OGBN-Arxiv |
|---|---|
| VGNN-entropy | 70.49±6.80 |
| VGNN-max-score | 69.82±6.30 |
| VGNN-dropout | 69.69±6.62 |
| VGNN-ensemble | 71.07±5.55 |
| VGNN-energy | 70.08±10.42 |
| VGNN-gnnsafe | 62.36±9.82 |
| SGNN-GKDE | 48.87±4.18 |
| EGNN | 64.12±8.74 |
| GEBM | 56.84±11.42 |
| EPN | 83.90±1.53 |
| X-EviProbe (Full set) | **89.61±1.87** |
| X-EviProbe (20K sampling) | 84.67±3.51 |

*Table 19.* OGBN-Arxiv Full Results (incl. Subsampling): Misclassification Detection. We report MIS_AUROC (↑) on `ogbn-arxiv`. The best performance is marked in **bold**.

| Model | OGBN-Arxiv |
|---|---|
| VGNN-entropy | 75.64±0.18 |
| VGNN-max-score | 77.61±0.13 |
| VGNN-dropout | 77.63±0.18 |
| VGNN-ensemble | **77.74±0.05** |
| VGNN-energy | 68.85±0.29 |
| VGNN-gnnsafe | 60.84±0.25 |
| SGNN-GKDE | 58.05±7.32 |
| EGNN | 77.09±0.35 |
| GEBM | 53.81±0.84 |
| EPN | 76.88±0.08 |
| X-EviProbe (Full set) | 74.98±0.23 |
| X-EviProbe (Full set w/o logits) | 77.67±0.13 |
| X-EviProbe (20K sampling w/o logits) | 77.14±0.16 |

## G.3. Runtime Comparison

Tables 20 and 21 report full-dataset wall-clock runtime on a single NVIDIA RTX 5090 GPU. Extra Train denotes additional training time beyond the shared frozen GCN backbone, while Inference (UQ) denotes the time required to obtain uncertainty scores over all test nodes.

*Table 20.* Full Runtime Comparison (Part I) — Wall-clock seconds for the full dataset on a single NVIDIA RTX 5090 GPU. **Extra Train (s)**: total additional training time beyond the shared frozen GCN backbone. **Inference (UQ, s)**: total wall-clock time for obtaining uncertainty scores at inference over all test nodes. **Total (s)**: sum of Extra Train and Inference, representing the complete additional cost to obtain uncertainty scores from a frozen GCN. VGNN-* methods derive UQ scores directly from GCN outputs with negligible overhead. Methods with Extra Train = 0 require no additional training.

| Dataset | Method | Extra Train (s) | Inference (UQ, s) | Total (s) |
|---|---|---|---|---|
| CoraML | VGNN-entropy | 0.00 | 0.03 | 0.03 |
| | VGNN-max-score | 0.00 | 0.03 | 0.03 |
| | VGNN-energy | 0.00 | 0.04 | 0.04 |
| | VGNN-gnnsafe | 0.00 | 0.03 | 0.03 |
| | MC Dropout | 0.00 | 0.19 | 0.19 |
| | Ensemble | 79.56 | 0.15 | 79.71 |
| | SGNN-GKDE | 10.49 | 0.07 | 10.56 |
| | EGNN | 5.02 | 0.06 | 5.08 |
| | EPN | 6.56 | 0.08 | 6.64 |
| | X-EviProbe | **0.00** | 0.24 | **0.24** |
| CiteSeer | VGNN-entropy | 0.00 | 0.04 | 0.04 |
| | VGNN-max-score | 0.00 | 0.03 | 0.03 |
| | VGNN-energy | 0.00 | 0.04 | 0.04 |
| | VGNN-gnnsafe | 0.00 | 0.04 | 0.04 |
| | MC Dropout | 0.00 | 0.19 | 0.19 |
| | Ensemble | 72.01 | 0.11 | 72.12 |
| | SGNN-GKDE | 10.39 | 0.07 | 10.46 |
| | EGNN | 8.45 | 0.06 | 8.51 |
| | EPN | 4.18 | 0.07 | 4.25 |
| | X-EviProbe | **0.00** | 0.33 | **0.33** |
| PubMed | VGNN-entropy | 0.00 | 0.07 | 0.07 |
| | VGNN-max-score | 0.00 | 0.07 | 0.07 |
| | VGNN-energy | 0.00 | 0.07 | 0.07 |
| | VGNN-gnnsafe | 0.00 | 0.07 | 0.07 |
| | MC Dropout | 0.00 | 0.73 | 0.73 |
| | Ensemble | 47.42 | 0.33 | 47.75 |
| | SGNN-GKDE | 8.71 | 0.15 | 8.86 |
| | EGNN | 4.08 | 0.12 | 4.19 |
| | EPN | 10.03 | 0.18 | 10.21 |
| | X-EviProbe | **0.00** | 4.79 | **4.79** |
| AmazonPhoto | VGNN-entropy | 0.00 | 0.05 | 0.05 |
| | VGNN-max-score | 0.00 | 0.05 | 0.05 |
| | VGNN-energy | 0.00 | 0.05 | 0.05 |
| | VGNN-gnnsafe | 0.00 | 0.06 | 0.06 |
| | MC Dropout | 0.00 | 0.72 | 0.72 |
| | Ensemble | 110.70 | 0.60 | 111.30 |
| | SGNN-GKDE | 15.23 | 0.15 | 15.38 |
| | EGNN | 6.97 | 0.11 | 7.08 |
| | EPN | 6.26 | 0.20 | 6.46 |
| | X-EviProbe | **0.00** | 0.99 | **0.99** |

*Table 21.* Full Runtime Comparison (Part II) — Continuation of Table 20.

| Dataset | Method | Extra Train (s) | Inference (UQ, s) | Total (s) |
|---|---|---|---|---|
| AmazonComputers | VGNN-entropy | 0.00 | 0.07 | 0.07 |
| | VGNN-max-score | 0.00 | 0.07 | 0.07 |
| | VGNN-energy | 0.00 | 0.07 | 0.07 |
| | VGNN-gnnsafe | 0.00 | 0.07 | 0.07 |
| | MC Dropout | 0.00 | 1.09 | 1.09 |
| | Ensemble | 76.74 | 0.81 | 77.55 |
| | SGNN-GKDE | 15.47 | 0.21 | 15.68 |
| | EGNN | 6.51 | 0.15 | 6.66 |
| | EPN | 4.04 | 0.23 | 4.27 |
| | X-EviProbe | **0.00** | 1.11 | **1.11** |
| CoauthorCS | VGNN-entropy | 0.00 | 0.07 | 0.07 |
| | VGNN-max-score | 0.00 | 0.07 | 0.07 |
| | VGNN-energy | 0.00 | 0.08 | 0.08 |
| | VGNN-gnnsafe | 0.00 | 0.08 | 0.08 |
| | MC Dropout | 0.00 | 0.94 | 0.94 |
| | Ensemble | 146.74 | 0.70 | 147.44 |
| | SGNN-GKDE | 23.96 | 0.20 | 24.16 |
| | EGNN | 7.52 | 0.14 | 7.67 |
| | EPN | 3.13 | 0.33 | 3.46 |
| | X-EviProbe | **0.00** | 4.46 | **4.46** |
| CoauthorPhysics | VGNN-entropy | 0.00 | 0.11 | 0.11 |
| | VGNN-max-score | 0.00 | 0.11 | 0.11 |
| | VGNN-energy | 0.00 | 0.11 | 0.11 |
| | VGNN-gnnsafe | 0.00 | 0.12 | 0.12 |
| | MC Dropout | 0.00 | 1.82 | 1.82 |
| | Ensemble | 88.21 | 1.29 | 89.50 |
| | SGNN-GKDE | 17.07 | 0.35 | 17.42 |
| | EGNN | 5.88 | 0.22 | 6.10 |
| | EPN | 6.98 | 0.73 | 7.71 |
| | X-EviProbe | **0.00** | 15.07 | **15.07** |
| OGBN-Arxiv | VGNN-entropy | 0.00 | 0.52 | 0.52 |
| | VGNN-max-score | 0.00 | 0.52 | 0.52 |
| | VGNN-energy | 0.00 | 0.52 | 0.52 |
| | VGNN-gnnsafe | 0.00 | 0.56 | 0.56 |
| | MC Dropout | 0.00 | 45.32 | 45.32 |
| | Ensemble | 835.63 | 15.77 | 851.40 |
| | SGNN-GKDE | 51.51 | 4.05 | 55.56 |
| | EGNN | 32.74 | 2.43 | 35.17 |
| | EPN | 3.14 | 4.16 | 7.30 |
| | X-EviProbe (20K) | **0.00** | 5.16 | **5.16** |
| | X-EviProbe (Full) | **0.00** | 45.25 | 45.25 |

## G.4. Generalization to GAT Backbones

For the GAT-backbone experiments, we follow the GAT setting of Yu et al. (Yu et al., 2025) and replace the frozen GCN backbone with a two-layer GAT backbone, while keeping the same post-hoc X-EviProbe procedure. Tables 22 and 23 report the corresponding OOD and misclassification detection results. These results verify that X-EviProbe is not tied to the GCN backbone, but only requires access to frozen node embeddings, logits, and probabilities.

*Table 22.* GAT Backbone — OOD Detection (AUROC ↑) across 7 datasets.

| Model | CoraML | CiteSeer | PubMed | Amazon Photo | Amazon Computers | Coauthor CS | Coauthor Physics |
|---|---|---|---|---|---|---|---|
| VGNN-entropy | 89.61±1.63 | 90.12±2.20 | 63.82±7.42 | 91.37±2.33 | 88.16±3.12 | 90.90±2.55 | 93.74±0.58 |
| VGNN-max-score | 88.65±1.68 | 89.60±2.19 | 63.82±7.42 | 90.53±2.92 | 85.53±3.33 | 89.77±2.89 | 93.02±1.07 |
| VGNN-dropout | 88.21±2.11 | 89.07±2.56 | 62.67±7.30 | 89.95±3.03 | 83.32±4.04 | 89.68±2.83 | 92.00±2.06 |
| VGNN-ensemble | 88.22±1.84 | 89.43±2.91 | 65.72±6.16 | 90.38±3.66 | 84.43±4.22 | 89.45±3.29 | 92.38±1.94 |
| VGNN-energy | 89.89±1.45 | 90.29±2.19 | 63.83±7.33 | 91.22±2.22 | 88.63±3.00 | 91.22±1.80 | 94.73±0.94 |
| VGNN-gnnsafe | 91.44±1.63 | 91.08±2.01 | 64.78±8.52 | 92.40±1.46 | 89.74±3.46 | 93.96±1.57 | 96.24±0.71 |
| SGNN-GKDE | 83.26±4.81 | 89.14±2.16 | 64.92±9.04 | 48.96±3.85 | 48.23±5.95 | 38.92±10.64 | 94.03±1.88 |
| EGNN | 88.62±1.23 | 88.86±1.41 | 64.03±9.93 | 88.32±3.81 | 84.17±4.60 | 90.82±2.63 | 93.35±1.97 |
| EPN | 89.84±1.78 | 90.24±1.77 | 67.07±1.65 | 91.14±2.43 | 86.98±3.63 | 93.11±2.09 | 94.07±2.05 |
| **X-EviProbe** | **95.14±1.39** | **98.15±0.52** | **86.53±4.32** | **97.20±0.45** | **95.18±1.13** | **98.75±0.23** | **98.87±0.46** |

*Table 23.* GAT Backbone — Misclassification Detection (MIS_AUROC ↑) across 7 datasets. We use strikethrough to indicate invalid low-accuracy results.

| Model | CoraML | CiteSeer | PubMed | Amazon Photo | Amazon Computers | Coauthor CS | Coauthor Physics |
|---|---|---|---|---|---|---|---|
| VGNN-entropy | 76.60±2.88 | 86.88±1.90 | 72.93±1.37 | 84.04±2.84 | 79.97±1.40 | 80.02±1.34 | 88.88±1.49 |
| VGNN-max-score | 77.94±2.97 | 87.17±1.94 | 73.54±1.29 | 84.79±2.43 | 82.65±1.27 | 82.95±1.01 | 89.36±1.31 |
| VGNN-dropout | 79.81±2.09 | 86.48±2.54 | 72.62±2.13 | 84.55±1.32 | 82.43±1.00 | 84.39±0.47 | 87.68±1.03 |
| VGNN-ensemble | 79.61±1.00 | 86.65±1.10 | 74.50±0.96 | 85.38±1.57 | 82.70±0.98 | 83.28±0.56 | 88.62±0.97 |
| VGNN-energy | 75.93±3.08 | 86.52±1.92 | 70.72±1.85 | 80.97±3.39 | 74.04±3.00 | 73.82±1.95 | 87.42±2.14 |
| VGNN-gnnsafe | 74.72±2.89 | 86.28±1.71 | 69.34±1.56 | 77.76±3.26 | 73.45±3.05 | 71.37±1.79 | 84.85±1.84 |
| SGNN-GKDE | ~~85.01±8.78~~ | 78.73±7.82 | 75.24±2.87 | 64.62±16.19 | 59.67±15.34 | 49.24±19.55 | 89.45±1.69 |
| EGNN | 77.55±2.62 | 86.99±1.82 | 75.71±1.85 | 85.36±1.68 | 80.26±2.13 | 83.37±1.71 | 89.49±1.37 |
| **X-EviProbe** | **83.15±1.98** | **92.12±1.38** | **79.73±0.89** | **88.74±1.24** | **86.32±1.37** | **84.97±1.03** | **91.73±0.76** |

## G.5. Stress Test under Distorted Latent Geometry

We include heterophilous graphs as an intentionally challenging stress test for X-EviProbe under poorly structured frozen representations. These results are not intended to benchmark node classification on heterophilous graphs; rather, they examine whether the post-hoc uncertainty signal remains useful when a standard GCN backbone has low predictive accuracy.

*Table 24.* Heterophilous Benchmarks — Misclassification Detection (MIS_AUROC (↑)) across 3 datasets. SGNN-GKDE and EGNN train independent models from scratch; all other baselines and X-EviProbe operate on the same frozen GCN. Under this GCN setting with only 20 training nodes per class, the test accuracy drops to 43.24%, 19.49%, and 28.32% on Chameleon, Actor, and Squirrel, respectively, simulating an extreme UQ scenario.

| Model | Chameleon | Actor | Squirrel |
|---|---|---|---|
| VGNN-entropy | 64.77±3.72 | 50.48±2.33 | 59.78±1.65 |
| VGNN-max-score | 65.81±3.92 | 50.61±2.42 | 59.57±1.60 |
| VGNN-dropout | 62.40±5.20 | 50.25±1.78 | 59.68±3.73 |
| VGNN-energy | 65.58±2.88 | 50.40±2.33 | 59.59±2.89 |
| VGNN-gnnsafe | 61.67±2.96 | 50.09±2.27 | 56.54±2.58 |
| SGNN-GKDE | **69.85±1.91** | 50.69±1.32 | 57.84±2.86 |
| EGNN | 68.66±3.85 | **53.12±2.22** | 55.95±2.00 |
| **X-EviProbe** | 69.07±3.76 | 50.31±2.40 | **61.19±1.82** |

*Table 25.* Heterophilous Benchmarks — OOD Detection (AUROC ↑) across 3 datasets.

| Model | Chameleon | Actor | Squirrel |
|---|---|---|---|
| VGNN-entropy | 54.91±9.34 | 50.20±1.33 | 45.99±11.50 |
| VGNN-max-score | 53.69±8.69 | 50.14±1.22 | 45.76±11.63 |
| VGNN-dropout | 56.75±7.51 | 49.76±1.29 | 47.63±11.30 |
| VGNN-energy | 56.52±8.85 | 50.42±1.14 | 46.05±10.19 |
| VGNN-gnnsafe | 54.08±5.64 | 50.22±0.69 | 48.05±7.00 |
| SGNN-GKDE | 56.25±3.90 | 49.98±1.19 | 49.63±13.20 |
| EGNN | 56.76±7.00 | 50.39±1.85 | 50.32±14.16 |
| **X-EviProbe** | **64.85±9.24** | **62.56±3.74** | **53.08±4.44** |

## G.6. Additional Component and Propagation Ablations

Table 26 compares scalar evidence-strength propagation with class-wise vector propagation. Table 27 provides an additive view of X-EviProbe by progressively introducing probability direction, geometric evidence, logit magnitude, and propagation.

Since propagation is linear, scalar and vector propagation are equivalent for epistemic uncertainty after summing over classes. Their difference mainly affects aleatoric uncertainty, where vector propagation mixes class-wise neighbor predictions while scalar propagation only diffuses total evidence strength. As shown in Table 26, vector propagation yields small gains on some datasets but is worse on others, and thus does not provide a consistent empirical advantage over scalar propagation. We therefore adopt scalar propagation as the default: it achieves comparable performance while propagating only the total evidence strength rather than class-wise evidential parameters, keeping the post-hoc step lightweight and avoiding class-level signal mixing from neighboring nodes.

Table 27 provides a complementary additive view of the framework. For OOD detection, performance improves mono-

tonically as geometry, logit magnitude, and propagation are progressively introduced; in particular, the Prob-only variant stays close to random because it assigns nearly identical epistemic strength to all nodes. For misclassification detection, geometry generally improves over Prob-only, while adding logit magnitude can partially reduce the gain. The w/o-logits variant therefore serves as a useful task-specific choice for misclassification detection, whereas the full X-EviProbe remains the unified default used in the main experiments.

*Table 26.* Scalar vs. Vector Propagation Comparison — Misclassification AUROC for both variants across 7 datasets.

| Model | CoraML | CiteSeer | PubMed | Amazon Photo | Amazon Computers | Coauthor CS | Coauthor Physics |
|---|---|---|---|---|---|---|---|
| X-EviProbe(vector) | 87.68±0.68 | 92.58±0.44 | 78.96±1.16 | 88.57±0.68 | 81.40±1.34 | 89.55±0.50 | 93.54±0.68 |
| X-EviProbe(scalar) | 87.65±0.73 | 92.07±0.54 | 79.33±1.21 | 88.67±1.08 | 83.86±1.22 | 88.18±0.65 | 92.53±0.63 |

*Table 27.* Additive Ablation Study. We progressively introduce components to examine their cumulative contributions. **Prob-only**: $\alpha = \hat{p} + 1$ (probability direction only, unit evidence). **+Geometry**: $\alpha = (\sum_c e_c^{\text{geo}}) \cdot \hat{p} + 1$ (geometric evidence modulates evidence strength). **+Geometry, Logits**: $\alpha = (\lambda \cdot \sum_c \text{softplus}(z_c)) \cdot \hat{p} + 1$ (geometry gates logit magnitude; equivalent to "w/o propagation" in Table 4). **+Propagation (Full)**: complete X-EviProbe with evidence strength propagation (Eq. 7). For misclassification detection, we additionally report **Full (w/o Logits)**, which removes the logit magnitude component from the full framework, as motivated by the consistent finding in Table 7 that logit magnitude is counterproductive for this task. Best in **bold**, runner-up underlined.

**(a) OOD Detection: AUROC (↑)**

| Variant | CoraML | CiteSeer | PubMed | Amazon Photo | Amazon Computers | Coauthor CS | Coauthor Physics | OGBN-Arxiv |
|---|---|---|---|---|---|---|---|---|
| Prob-only | 50.00 | 50.05 | 50.00 | 49.92 | 49.95 | 49.94 | 50.32 | 49.96 |
| +Geometry | 66.13 | 69.68 | 57.59 | 87.38 | 80.50 | 88.98 | 84.92 | 77.83 |
| +Geometry, Logits | 78.87 | 81.55 | 60.76 | 90.55 | 84.16 | 91.79 | 90.28 | 80.31 |
| +Propagation (Full) | **94.52** | **98.19** | **83.30** | **97.08** | **93.35** | **98.91** | **97.99** | **89.61** |

**(b) Misclassification Detection: AUROC (↑)**

| Variant | CoraML | CiteSeer | PubMed | Amazon Photo | Amazon Computers | Coauthor CS | Coauthor Physics | OGBN-Arxiv |
|---|---|---|---|---|---|---|---|---|
| Prob-only | 83.76 | 84.67 | 75.15 | 86.01 | 74.63 | 87.26 | 88.52 | 77.59 |
| +Geometry | 85.04 | 90.54 | 74.41 | 87.69 | 76.71 | 87.97 | 90.36 | 75.69 |
| +Geometry, Logits | 84.39 | 90.09 | 74.02 | 84.88 | 74.48 | 85.26 | 88.92 | 73.43 |
| +Propagation (Full) | 87.65 | 92.07 | 79.33 | 88.67 | 83.86 | 88.18 | 92.53 | 74.98 |
| Full (w/o Logits) | **87.66** | **92.74** | **79.99** | **90.07** | **84.48** | **89.54** | **93.41** | **77.67** |

