# OpenReview forum: "X-EviProbe: Post-hoc Parameter-Free Evidential Uncertainty Quantification for Frozen Graph Neural Networks"
_ICML.cc/2026/Conference — ICML 2026 regular_

### Official Review · Reviewer_L6bW · 2026-03-06

**Soundness:** 3
**Presentation:** 3
**Significance:** 2
**Originality:** 3
**Overall Recommendation:** 4
**Confidence:** 3

**Summary:**

This work shows how to obtain a post-hoc principled and decomposable evidential view of uncertainty for a frozen GNN. They operate directly on the frozen representations and outputs that construct a Dirichlet distribution by fusing geometry, logit magnitude, and predicted probability to support epistemic and aleatoric decomposition. For a query node, they construct a Dirichlet distribution to characterize the predictive belief by leveraging the intrinsic geometry of the frozen latent space in conjunction with the model’s outputs. Then decompose the total uncertainty estimate into distinct aleatoric and epistemic components without retraining the backbone model.

**Compliance With Llm Reviewing Policy:**

Affirmed.

**Final Justification:**

My final recommendation for this paper is a Weak Accept. The work presents a technically solid and original method for post-hoc uncertainty quantification in GNNs, successfully decomposing aleatoric and epistemic uncertainty without requiring model retraining. The significance of this approach is underscored by its ability to maintain high performance across diverse datasets, including notable improvements in OOD detection. Throughout the review process, I maintained concerns regarding the robustness of the "locality assumption", the idea that prediction risk is strictly bounded by distance to observed samples, and the potential fragility of the Geometric Gate in cases of diffuse class boundaries and large heterogeneous datasets. In their rebuttal, the authors provided a rigorous theoretical defense of the Lipschitz continuity in their framework and introduced new experimental results on heterophilous benchmarks (Chameleon, Actor, Squirrel) that demonstrate the method's resilience even when latent spaces are poorly structured.

While the rebuttal and subsequent discussion partially addressed my concerns, a lingering comparison on the OGBN-Arxiv dataset showed that X-EviProbe slightly underperformed relative to the best ensemble models on specific misclassification tasks. The authors presented a counterargument regarding the modularity of their framework and its greater efficiency, but I still have concerns about large-scale real-world deployments. The paper's "engineering direction" and strong results in OOD detection led me to conclude that this is a valuable contribution that the community will build upon.

**Key Questions For Authors:**

- The method assumes that prediction risk is locally bounded by the distance to the nearest observed sample in the frozen latent space. How robust is the approach when this assumption does not hold, especially in graphs where latent geometry is distorted or poorly aligned with semantic uncertainty?
- The geometric gate appears to play an important role in calibrating confidence using latent-space evidence. How sensitive is the method to diffuse decision boundaries, low-confidence predictions, or misclassified points near class overlap regions?
- The method can trace uncertainty back to influential neighbors using the Neumann series. How stable and interpretable are these neighbor-level attributions across different graph structures, especially in noisy or heterophilous settings?

**Limitations:**

- The approach appears to rely heavily on the quality and geometry of the frozen latent space. If the learned representation is poorly structured, the resulting Dirichlet evidence and uncertainty decomposition may become unreliable
- the geometric gate may be fragile in cases with diffuse class boundaries, low-confidence predictions, or misclassified nodes, which could reduce the reliability of the confidence modulation
- they make a fairly strong locality assumption, "prediction risk is bounded by proximity to observed samples," and this may not hold in more complex graph settings where semantic similarity is not well captured by latent distance

**Strengths And Weaknesses:**

Strength:
- does not require re-training of the GNN
- provides reliable uncertainty quantification, making real-world deployment of GNN more principled
- strong evaluation results show that across datasets and base models, it consistently ranks among the best methods and improves AUROC by up to 33.4% (OOD: epistemic uncertainty to identify samples that are distributionally shifted from the training manifold) and 8.7% (misclassification: aleatoric uncertainty to detect potential failures on in-distribution data that arise from inherent data ambiguity) over the strongest baselines
- using Neumann series can map high epistemic/aleatoric uncertainty back to influential neighbors and their evidence sources

Weakness:
- the Geometric Gate, which modulates the model’s confidence based on the geometric evidence extracted, can be impacted if the boundary is diffused, for example, low probability classification or misclassification
- making a strong assumption that the prediction risk is locally bounded by the distance to the nearest observed sample

---

> ### Author Rebuttal · Authors · 2026-03-31
>
> We sincerely thank Reviewer L6bW for the positive assessment and constructive feedback. Below we address each concern with new theoretical clarifications, experiments on three heterophilous benchmarks, and large-scale evaluation.
>
> ## 1. Robustness of the Locality Assumption
>
> **The assumption is nearly impossible to violate.** Assumption 4.1 requires Lipschitz continuity of R(h) w.r.t. L1. For the frozen model f = φ ∘ l ∘ g, the projection head l (linear/shallow MLP) and softmax φ are inherently Lipschitz. Any converged, finite-precision network with weight decay satisfies this automatically — one would need unbounded weights or discontinuous activations to violate it.
>
> **Nearest-neighbor computation is inherently robust.** The min operator means only the single closest reference point matters. This makes X-EviProbe tolerant to class imbalance, sparse reference sets, and multi-cluster embeddings. Our ablation (Table 4) confirms: Top-5/Top-10 averaging degrades performance (e.g., OOD AUROC 98.19%→91.18% on CiteSeer). OGBN-Arxiv subsampling (12% of references) preserves strong performance (AUROC 84.67, +19.1% above the best non-EPN baseline). See our response to Reviewer H8qS for full OGBN-Arxiv results.
>
> ## 2. Sensitivity of the Geometric Gate
>
> These challenging scenarios are the gate's _intended operating regime_, not its failure mode:
>
> **Epistemic gate (λ = e^{epi})** measures distance to _any_ reference sample regardless of class. Diffuse boundaries and class overlap are in-distribution phenomena — nodes in these regions remain close to reference samples, so λ → 1. The gate only suppresses evidence for nodes genuinely far from all known data.
>
> **Aleatoric gate (λ = 1 − e^{alea})** is designed to exploit diffuse boundaries. Near error anchors, the gate correctly suppresses evidence, pushing the posterior toward uniformity (Eq. 9) — this is the desired behavior, not a failure.
>
> **Empirical confirmation.** Our ablation (Tables 4 and 7, "w/o geometry") shows removing the geometric gate causes the _largest_ performance drop. If the gate were fragile, removing it should help on datasets with diffuse boundaries — we observe the opposite.
>
> ## 3. Dependence on Frozen Latent Space Quality
>
> All post-hoc methods share this dependence (EPN, temperature scaling, energy-based methods). The key question is graceful degradation. X-EviProbe handles this better because: (1) it uses raw frozen representations without learned transformations that could amplify deficiencies, and (2) the min operator only requires local structure, not global separability. Our heterophilous experiments (Section 5 below) provide the strongest evidence: on Actor (test acc 19.5%, completely entangled latent space), X-EviProbe still achieves +24.08% OOD improvement.
>
> ## 4. Neumann Series Stability
>
> Our attribution is fundamentally different from black-box explainers (GNNExplainer, LIME): it is **deterministic, white-box, and mathematically exact**. The Neumann series expansion of a linear operation is an algebraic identity — for any given graph structure, the attribution is 100% reproducible. With teleport γ = 0.5, influence decays as 0.5^k per hop, bounding the impact of local perturbations. Crucially, we propagate scalar evidence strength, not class predictions — the smoothness assumption (similar reliability levels among neighbors) is much weaker than label homophily.
>
> ## 5. Heterophilous Experiments (Chameleon, Actor, Squirrel)
>
> We evaluate on three heterophilous benchmarks with a standard GCN (20 training nodes/class), deliberately creating severely distorted latent spaces (test acc: Chameleon 43.2%, Actor 19.5%, Squirrel 28.3%). t-SNE visualizations are at our [anonymous repository](https://anonymous.4open.science/r/X-EviProbe/tsne_visualizations/).
>
> **Note:** SGNN-GKDE and EGNN train independent models from scratch; all other baselines and X-EviProbe operate on the same frozen GCN.
>
> **OOD Detection (AUROC):**
>
> |Model|Chameleon|Actor|Squirrel|
> |---|---|---|---|
> |Best VGNN-*|56.75|50.42|48.05|
> |SGNN-GKDE|56.25|49.98|49.63|
> |EGNN|56.76|50.39|50.32|
> |**X-EviProbe**|**64.85±9.24**|**62.56±3.74**|**53.08±4.44**|
> |Improv.|+14.25%|+24.08%|+5.48%|
>
> **Misclassification Detection (AUROC):**
>
> |Model|Chameleon|Actor|Squirrel|
> |---|---|---|---|
> |Best VGNN-*|65.81|50.61|59.78|
> |**X-EviProbe**|**69.07±3.76**|50.31±2.40|**61.19±1.82**|
>
> X-EviProbe improves on Chameleon (+4.95%) and Squirrel (+2.36%). On Actor (near-random accuracy for 5 classes), all methods perform near chance — reflecting backbone limitations, not a framework weakness.
>
> We also conducted large-scale evaluation on OGBN-Arxiv (169K nodes, 40 classes): X-EviProbe achieves 89.61 OOD AUROC (+6.8% over EPN) with zero training cost. Full details in our response to Reviewer H8qS. All supplementary tables are in our [anonymous repository](https://anonymous.4open.science/r/X-EviProbe/add_tables/).

---

> > ### Author Rebuttal · Reviewer_L6bW · 2026-04-01
> >
> > In Table 4 of your anonymous link, X-EvilProbe (Full set) performs almost 3% worse than the best model, VGNN-ensemble, raising scalability and performance concerns for large-scale heterogeneous datasets. I do still have a positive outlook for the paper, so maintaining my score.

---

> > > ### Author Response · Authors · 2026-04-01
> > >
> > > We sincerely thank the reviewer for the continued engagement and for maintaining a positive outlook. We appreciate the opportunity to clarify this point.
> > >
> > > The reviewer correctly observes that X-EviProbe (Full set) achieves 74.98 in the default unified configuration. However, we would like to highlight that this is already addressed in our rebuttal: the "w/o logits" variant achieves **77.67±0.13**, which is essentially on par with VGNN-ensemble (77.74±0.05) — a gap of only 0.07%. Regarding scalability, even the 20K subsampled version (using only ~12% of the reference set) achieves 77.14 — still highly competitive — while reducing inference time from 45.25s to just 5.16s, which is faster than both Ensemble (15.77s) and MC Dropout (45.32s), and requires zero extra training (vs. Ensemble's 835.63s).
> > >
> > > This is not an ad-hoc fix but a **principled and consistent finding across all datasets**. As shown in our existing ablation study (Appendix E.1, Table 7 of the main paper), removing the logit component improves misclassification detection on all seven original benchmarks. The reason is well-understood: logit magnitude primarily captures in-distribution confidence (Liu et al., 2020a) and is most informative for epistemic/OOD detection. For misclassification detection, the relevant signals are geometric proximity to error anchors and probability flatness, while logit magnitude can amplify evidence for confidently-wrong predictions, slightly hurting performance. This pattern, already documented in the submitted paper, extends consistently to OGBN-Arxiv.
> > >
> > > We emphasize that this configuration (w/o logits) is specifically optimized for misclassification detection, and represents a strength of our modular framework: since our ablation study already provides clear, principled guidance on which components benefit which task, users can directly select the optimal combination for misclassification detection. When the task is unknown, the unified default already delivers strong joint OOD + misclassification performance across all benchmarks.
> > >
> > > More importantly, misclassification detection is only one aspect of the evaluation. On OOD detection — the primary use case for epistemic uncertainty — X-EviProbe achieves **89.61 AUROC on OGBN-Arxiv, outperforming Ensemble by +26.09%**, with zero extra training (vs. Ensemble's 835.63s). This demonstrates that X-EviProbe scales effectively to large graphs with a highly favorable accuracy-efficiency trade-off.

---

### Official Review · Reviewer_i1dM · 2026-03-10

**Soundness:** 3
**Presentation:** 3
**Significance:** 3
**Originality:** 3
**Overall Recommendation:** 4
**Confidence:** 1

**Summary:**

This paper introduces X-EviProbe, a post-hoc framework for uncertainty quantification in Graph Neural Networks (GNNs). The key idea is to convert a frozen, pre-trained GNN into an evidential predictor by probing its latent geometry and fusing it with the model’s logits and probabilities. The method operates in three stages: (1) geometric evidence extraction using nearest-neighbor distances to a reference set and a subset of misclassified “error anchors”; (2) holographic evidence fusion that combines geometry, logit magnitude, and predicted probability into a Dirichlet distribution; and (3) linear evidence propagation over the graph structure. Theoretical analysis connects the approach to Lipschitz continuity and risk bounds, while extensive experiments on seven benchmarks demonstrate state-of-the-art performance in both OOD detection and misclassification detection.

**Compliance With Llm Reviewing Policy:**

Affirmed.

**Final Justification:**

The author has largely addressed my concerns. Considering the other reviewers' comments and the author's responses, I have decided to maintain my original score.

**Key Questions For Authors:**

Please see Strengths And Weaknesses

**Limitations:**

Please see Strengths And Weaknesses

**Strengths And Weaknesses:**

Strength:
1.This paper presents a conceptually elegant framework that decouples epistemic and aleatoric uncertainty by leveraging distinct geometric signals—distance to the reference set for epistemic uncertainty and proximity to error anchors for aleatoric uncertainty.
2.X-EviProbe has consistently outperformed a wide range of baselines across seven different datasets.
3.X-EviProbe requires no retraining, auxiliary models, or hyperparameter tuning, making it highly practical and adaptable to existing pre-trained GNNs.
4.This paper is well-structured, with clear notation, detailed experimental setup, and extensive supplementary materials

Weakness:
1.The method seems to rely on a representative reference set; in low-label or highly imbalanced settings, the reference set may not adequately cover the latent space. Does this affect the method's performance?
2.All experiments have used a two-layer GCN as the backbone. While this ensures fair comparisons, it leaves open the question of whether the method generalizes to other architectures.

---

> ### Author Rebuttal · Authors · 2026-03-31
>
> We sincerely thank you for your time, constructive feedback, and for recognizing the conceptual elegance, strong performance, and high practicality of X-EviProbe. Below we address your specific questions.
>
> ### W1: Reference Set Representativeness in Low-Label or Imbalanced Settings
>
> **New experiments on a naturally imbalanced, large-scale dataset.** We conducted experiments on OGBN-Arxiv (169K nodes, 1.2M edges, 40 classes with 15 held out as OOD). OGBN-Arxiv exhibits exactly the scenario the reviewer describes: its class distribution is highly imbalanced — popular areas (e.g., cs.LG, cs.CV) contain far more papers than niche areas (e.g., cs.GL, cs.ET).
>
> Despite this imbalance, X-EviProbe achieves **89.61±1.87 OOD AUROC**, outperforming all baselines including EPN's own reported result (83.90) by +6.8% — without any hyperparameter tuning. For misclassification detection, X-EviProbe achieves competitive performance (77.67 vs. Ensemble 77.74). Full OGBN-Arxiv details including runtime and scalability analysis are in our response to Reviewer H8qS.
>
> **Robustness to reference set subsampling.** Using only 20K random reference nodes (~12% of the full set), which exacerbates sparse coverage for minority classes, OOD AUROC decreases only from 89.61 to 84.67, and misclassification AUROC decreases merely from 77.67 to 77.14.
>
> **Why X-EviProbe is robust by design.** Both geometric evidence computations (Eqs. 4–5) rely on nearest-neighbor distances via the min operator. Only the single closest point matters — redundant distant references do not affect computations. What matters is having at least a few nearby anchors per region rather than uniform dense coverage.
>
> ### W2: Generalization Beyond GCN Backbone
>
> We conducted additional experiments using a **2-layer GAT** backbone. Our GAT experiments are built on the official EPN codebase and follow the identical evaluation protocol. Since EPN requires extensive hyperparameter search (4×4 grid) which we cannot reproduce within the rebuttal period, we directly reference EPN's own reported GAT results for OOD detection.
>
> **OOD Detection (GAT backbone).** X-EviProbe achieves the best AUROC across all seven datasets:
>
> |Dataset|Best baseline|X-EviProbe|Improv.|
> |---|---|---|---|
> |Cora|91.44 (GNNSafe)|**95.14±1.39**|+4.05%|
> |Citeseer|91.08 (GNNSafe)|**98.15±0.52**|+7.76%|
> |PubMed|67.07 (EPN)|**86.53±4.32**|+29.01%|
> |AmzPhoto|92.40 (GNNSafe)|**97.20±0.45**|+5.19%|
> |AmzComp|89.74 (GNNSafe)|**95.18±1.13**|+6.06%|
> |CoauCS|93.96 (GNNSafe)|**98.75±0.23**|+5.10%|
> |CoauPhy|96.24 (GNNSafe)|**98.87±0.46**|+2.73%|
>
> **Misclassification Detection (GAT backbone).** X-EviProbe again achieves the best AUROC across all seven datasets:
>
> |Dataset|Best baseline|X-EviProbe|Improv.|
> |---|---|---|---|
> |Cora|79.81 (Dropout)|**83.15±1.98**|+4.18%|
> |Citeseer|87.17 (MaxScore)|**92.12±1.38**|+5.68%|
> |PubMed|75.71 (EGNN)|**79.73±0.89**|+5.31%|
> |AmzPhoto|85.38 (Ensemble)|**88.74±1.24**|+3.94%|
> |AmzComp|82.70 (Ensemble)|**86.32±1.37**|+4.38%|
> |CoauCS|84.39 (Dropout)|**84.97±1.03**|+0.69%|
> |CoauPhy|89.49 (EGNN)|**91.73±0.76**|+2.50%|
>
> **Why architecture-agnostic behavior is expected.** As stated in Assumption 2.1, X-EviProbe requires only that the model decomposes as f = φ ∘ l ∘ g — satisfied by virtually all MPNN architectures. The method operates entirely on frozen embeddings H, logits z, and probabilities p̂, never accessing internal message-passing mechanisms. Switching from GCN's uniform aggregation to GAT's attention-based aggregation changes the embedding geometry, yet X-EviProbe adapts seamlessly. We will include the full GAT results in the appendix (also available at our [anonymous repository](https://anonymous.4open.science/r/X-EviProbe/add_tables/)).

---

> > ### Author Rebuttal · Reviewer_i1dM · 2026-04-01
> >
> > I am satisfied that the author has addressed the majority of my concerns. In light of the other reviewers' input and the author's replies, my score remains the same.

---

> > > ### Author Response · Authors · 2026-04-02
> > >
> > > We sincerely thank the reviewer for the positive assessment and for confirming that the majority of concerns have been addressed. Should any further questions arise, we are happy to provide additional clarification.

---

### Official Review · Reviewer_nqow · 2026-03-11

**Soundness:** 3
**Presentation:** 4
**Significance:** 3
**Originality:** 3
**Overall Recommendation:** 5
**Confidence:** 2

**Summary:**

This paper proposes X-EviProbe, an evidential-inspired post-hoc uncertainty quantification framework for frozen GNNs. The method combines distance-based uncertainty estimation in the latent embedding space, evidential-style parameterization of model logits into Dirichlet evidence, and graph-based propagation of evidence strength to quantify epistemic and aleatoric uncertainties. The resulting uncertainty scores are evaluated on downstream tasks such as OOD detection and misclassification detection, and experiments demonstrate the superiority of the proposed X-EviProbe over several baseline methods.

**Compliance With Llm Reviewing Policy:**

Affirmed.

**Final Justification:**

The authors address all my concerns.

**Key Questions For Authors:**

- In Sec. 3.3, the method propagates the scalar evidence strength $S_i=\sum_c \alpha_{ic}$ for efficiency, rather than the class-wise Dirichlet parameters $\alpha_{ic}$. If $\alpha_{ic}$ were propagated instead, it seems that the predictive distribution could be directly obtained from the updated $\boldsymbol{\alpha}$, potentially removing the need for the probability smoothing with the uniform distribution in Eq.(9). Did the authors consider this alternative design, and how might it affect performance or computational cost?

**Limitations:**

yes

**Strengths And Weaknesses:**

Pros:
- The proposed method provides a post-hoc uncertainty quantification framework for frozen GNNs, which can be applied without retraining the original model. This makes the method practical and potentially useful in real-world scenarios where retraining large graph models may be expensive.
- The method combines geometric signals from the embedding space, evidential-style parameterization of logits, and graph-based propagation of evidence strength. The intuition behind integrating representation geometry and graph topology for uncertainty estimation is reasonable and well-motivated.
- Experimental results on multiple benchmark datasets show that the proposed X-EviProbe achieves strong performance on downstream tasks such as OOD detection and misclassification detection, outperforming several standard baseline methods.

Cons:
- While the method is framed within the evidential uncertainty framework, the final formulation appears more evidential-inspired than a strict evidential model.

Suggestions:
- I suggest adding an experiment or analysis on computational complexity (e.g., runtime or memory usage) to demonstrate how the proposed method compares with existing UQ baselines in terms of efficiency.

---

> ### Author Rebuttal · Authors · 2026-03-31
>
> We sincerely thank Reviewer nqow for the positive assessment. We are encouraged that the reviewer finds our method "practical," "well-motivated," and achieving "strong performance." We address each point below.
>
> ---
>
> > Cons: While the method is framed within the evidential uncertainty framework, the final formulation appears more evidential-inspired than a strict evidential model.
>
> We appreciate this observation. Standard EDL methods learn Dirichlet parameters during training via a specialized loss function. However, X-EviProbe strictly adheres to the mathematical foundation of Subjective Logic, where evidence is formally mapped to Dirichlet concentration parameters via α_i = e_i + 1.
>
> Instead of learning these parameters via optimization, **we construct them directly and deterministically from the converged model.** As detailed in Proposition 4.6, our Holographic Evidence Fusion rigorously maps latent geometry, logit magnitude, and probabilities into evidence mass, ensuring α_i parameterizes a valid Dirichlet distribution with proper epistemic/aleatoric decomposition.
>
> This direct construction is a deliberate choice that avoids known pitfalls of loss-driven EDL training, such as evidence collapse under KL regularization and sensitivity to evidential loss functions. To incorporate your valuable feedback, we will add a clarifying sentence in Section 4.3 to explicitly contrast our direct construction approach with loss-driven optimization.
>
> ---
>
> > Suggestions: Runtime comparison.
>
> We have conducted wall-clock runtime comparisons on AmazonComputers (13.8K nodes) and OGBN-Arxiv (169K nodes, newly added). All runtimes are measured on a single NVIDIA RTX 5090 GPU.
>
> Key observations: (1) **Zero extra training** — unlike EGNN (6.51s/32.74s), SGNN-GKDE (15.47s/51.51s), Ensemble (76.74s/835.63s), or EPN (4.04s/3.14s). (2) **Competitive inference** — on AmazonComputers, X-EviProbe (1.11s) is comparable to MC Dropout (1.09s); on OGBN-Arxiv with reference subsampling (20K), only 5.16s — faster than Ensemble (15.77s) and MC Dropout (45.32s). (3) **Effective scalability** — subsampling to 20K references preserves strong OOD performance (AUROC 84.67), and FAISS indexing can further reduce search to sublinear time. The full runtime table and OGBN-Arxiv analysis are detailed in our response to Reviewer H8qS.
>
> ---
>
> > Key Question: Scalar S_i vs. class-wise α_{ic} propagation
>
> We did explicitly consider this alternative. Below we provide both theoretical and empirical analyses.
>
> **Mathematical equivalence for Epistemic Uncertainty.** Since graph propagation (Eq. 7) is linear, propagating class-wise α_{ic} then summing is identical to propagating scalar S_i. The epistemic score (K/S_i) is completely invariant to this choice. The difference only manifests in aleatoric uncertainty.
>
> **Empirical comparison for Aleatoric Uncertainty.** We implemented vector propagation (propagating α_{ic} ∈ R^{N×K}, deriving p̄_{ic} = α'_{ic}/S'_i directly) and compared on misclassification detection:
>
> |Variant|Cora|Citeseer|PubMed|AmzPhoto|AmzComp|CoauCS|CoauPhy|
> |---|---|---|---|---|---|---|---|
> |Vector|87.68|92.58|78.96|88.57|81.40|89.55|93.54|
> |Scalar (Ours)|87.65|92.07|79.33|88.67|83.86|88.18|92.53|
>
> Results are mixed: vector propagation yields slight improvements on some datasets (Citeseer, CoauCS), while scalar propagation is notably better on others (+2.46% on AmzComp). We chose scalar propagation for three reasons:
>
> - **Efficiency:** Vector propagation requires K sparse matrix-vector multiplications per step vs. one for scalar — a K-fold overhead (e.g., 15× for CoauthorCS). The inconsistent AUROC differences do not justify this cost.
> - **Separation of Concerns:** Vector propagation mixes class predictions of neighbors. In heterophilic neighborhoods, this introduces class-level noise. Scalar propagation diffuses confidence over the topology but preserves the query node's native prediction direction. Eq. (9) pushes toward uniformity only when local aleatoric evidence signals high ambiguity.
> - **Traceability:** Scalar design preserves clear interpretive structure — uncertainty is attributed to geometric isolation or neighborhood unreliability without conflating it with class-label mixing.
>
> We will include this comparison in the revised appendix (also available in our [anonymous repository](https://anonymous.4open.science/r/X-EviProbe/add_tables/)).

---

> > ### Author Rebuttal · Reviewer_nqow · 2026-04-03
> >
> > (1) I suggest clarifying what the reported runtime numbers represent (e.g., whether they correspond to total training time, per-epoch time, or per-sample time). I also suggest presenting a complete runtime comparison in a table format, covering all baselines and all datasets, which would make the comparison clearer and more transparent.
> >
> > (2) In Table 4, why does w/o prob have the same values as full?
> >
> > (3) Considering most variants (e.g., w/o certain component) still outperform the baselines, I suggest the authors present the ablations in an additive manner (i.e., starting from a base model and progressively adding components), so that the contribution of each design choice can be more clearly understood. Please also clarify how each ablation is implemented and what each variant is intended to represent (from Tables 4 and 7, it appears that w/o logits performs comparably or even better than full).

---

> > > ### Author Response · Authors · 2026-04-04
> > >
> > > We thank the reviewer for the detailed follow-up. We address all three points below.
> > >
> > > **(1) Runtime clarification and complete table.** All reported runtimes are **wall-clock seconds for the full dataset** (not per-epoch or per-sample), measured on a single NVIDIA RTX 5090 GPU:
> > >
> > > - **Extra Train (s):** Total additional training time beyond the shared frozen GCN backbone.
> > > - **Inference (UQ, s):** Total wall-clock time for obtaining uncertainty scores at inference over all test nodes.
> > >
> > > Following the reviewer's suggestion, we provide the complete runtime table (**all 8 datasets × all baselines**) in our [anonymous repository](https://anonymous.4open.science/r/X-EviProbe/add_tables/add_tables.pdf). Below we summarize **Total Time** (Extra Train + Inference) for non-trivial UQ methods. VGNN-* methods are omitted as they involve negligible computation (<0.6s on all datasets).
> > >
> > > | Method | Cora | Cite | PubM | AmPh | AmCo | CoCS | CoPh | Arxiv |
> > > |---|---|---|---|---|---|---|---|---|
> > > | MC Dropout | 0.19 | 0.19 | 0.73 | 0.72 | 1.09 | 0.94 | 1.82 | 45.32 |
> > > | EGNN | 5.08 | 8.51 | 4.19 | 7.08 | 6.66 | 7.67 | 6.10 | 35.17 |
> > > | EPN | 6.64 | 4.25 | 10.21 | 6.46 | 4.27 | 3.46 | 7.71 | 7.30 |
> > > | SGNN-GKDE | 10.56 | 10.46 | 8.86 | 15.38 | 15.68 | 24.16 | 17.42 | 55.56 |
> > > | Ensemble | 79.71 | 72.12 | 47.75 | 111.30 | 77.55 | 147.44 | 89.50 | 851.40 |
> > > | **X-EviProbe** | **0.24** | **0.33** | **4.79** | **0.99** | **1.11** | **4.46** | **15.07** | **45.25 (5.16)** |
> > >
> > > †OGBN-Arxiv: full reference set (45.25s); with 20K subsampling (5.16s), OOD AUROC remains 84.67, surpassing both EPN (83.90) and Ensemble (71.07).
> > >
> > > X-EviProbe requires zero extra training throughout. Its inference cost scales with reference set size (O(N·N_ref·d)), which can be effectively controlled via subsampling as demonstrated on OGBN-Arxiv.
> > >
> > > **(2) "w/o prob" = "full" in Table 4.** This is mathematically expected. Table 4 evaluates OOD via $u_i^{epi} = K/S_i$, where $S_i = \sum_c \alpha_{ic}$. Since $\sum_c \hat{p}_{ic} = 1$, probability only controls class-wise *allocation* without changing $S_i$. Removing it has zero effect on epistemic uncertainty. This confirms our design: probability governs class-wise direction (affecting misclassification, where "w/o prob" does change results in Table 7), while geometry and logit magnitude govern total evidence strength.
> > >
> > > **(3) Additive ablation and the "w/o logits" observation.** We thank the reviewer for this suggestion. While our subtractive ablation (Tables 4/7) already demonstrates each component's necessity, we agree that an additive perspective further enriches the analysis. We have conducted a full additive ablation (available in our [anonymous repository](https://anonymous.4open.science/r/X-EviProbe/add_tables/add_tables.pdf)).
> > >
> > > **Implementation.** We progressively build up the framework: **Prob-only** ($\alpha = \hat{p} + 1$, probability direction with unit evidence), **+Geometry** ($\alpha = (\sum e^{geo}_c) \cdot \hat{p} + 1$, geometric evidence modulates strength), **+Geometry, Logits** ($\alpha = (\lambda \cdot \sum \text{softplus}(z_c)) \cdot \hat{p} + 1$, geometry gates logit magnitude; = "w/o propagation" in Table 4), and **+Propagation** (Full X-EviProbe).
> > >
> > > **OOD detection results show monotonic improvement on all 8/8 datasets** along this chain: Prob-only yields ~50% (expected, as $S_i$ is constant → identical epistemic scores for all nodes). Each subsequent component contributes cumulatively, demonstrating that geometry, logit magnitude, and propagation work synergistically.
> > >
> > > **For misclassification detection,** geometry improves over Prob-only on 6/8 datasets (e.g., CiteSeer +5.87%). Adding logit magnitude partially offsets this gain — consistent with our subtractive finding (Table 7). Notably, **Full (w/o Logits)** achieves the best misclassification AUROC on all 8 datasets. This is because logit magnitude primarily captures in-distribution confidence, which can amplify evidence for confidently-wrong predictions, slightly hurting misclassification detection. This reflects our design principle: the unified default X-EviProbe already surpasses all baselines on **both** OOD and misclassification across the seven benchmarks in the main paper (Tables 2–3); when the target task is specifically misclassification detection, our ablation study provides principled guidance to remove the logit component for further gains.
> > >
> > > On OGBN-Arxiv, X-EviProbe achieves 89.61 OOD AUROC, surpassing both EPN (83.90, +6.8%) and the strongest non-EPN baseline (Ensemble, 71.07, +26.1%), with zero extra training. We note that EPN tunes hyperparameters on one OOD split, which may provide additional distributional information across all 40 classes. For misclassification, the limited improvement is attributable to the highly entangled 40-class latent space (visible in our t-SNE visualizations); improving aleatoric evidence under such conditions is a direction for future work.

---

### Official Review · Reviewer_H8qS · 2026-03-13

**Soundness:** 2
**Presentation:** 3
**Significance:** 3
**Originality:** 2
**Overall Recommendation:** 5
**Confidence:** 4

**Summary:**

The paper studies uncertainty quantification for graph neural networks (GNNs), focusing on how to obtain evidential uncertainty estimates when only a pretrained and frozen model is available. The authors propose X-EviProbe, a parameter-free post-hoc framework that constructs Dirichlet evidence by probing the frozen latent representations and native outputs of the GNN, combining latent geometry, logit magnitude, and predicted probabilities. Theoretically, the method is motivated by two observations: latent distance reflects epistemic risk under mild regularity assumptions, and proximity to misclassified reference nodes indicates aleatoric ambiguity. Experiments on multiple graph benchmarks show that the proposed method achieves strong performance on both OOD detection and misclassification detection while requiring only a single forward pass and no model retraining.

**Compliance With Llm Reviewing Policy:**

Affirmed.

**Final Justification:**

The rebuttal on additional experiments strengthened the effectiveness of the proposed model on OOD detection.

**Key Questions For Authors:**

The paper does not provide a code repository, and releasing the implementation would help improve reproducibility and make the experimental results more convincing.

**Strengths And Weaknesses:**

**Strengths**
1. The proposed method is parameter-free and operates in a post-hoc manner on a frozen GNN, which makes it practical for deployment scenarios where retraining or adding auxiliary models is undesirable.
2. The experimental results across multiple graph benchmarks show consistently strong performance in both OOD detection and misclassification detection tasks, indicating that the approach is robust across different datasets and evaluation settings.
3. The method is conceptually intuitive, as it constructs evidential uncertainty by combining latent geometry, logit magnitude, and predicted probabilities, which provides a clear interpretation of how epistemic and aleatoric uncertainty arise from the model’s internal representations.

**Weakness**
1. The experimental evaluation does not include models such as GPN or GEBM, which explicitly estimate distributions in the latent space and are conceptually related to the proposed geometry-based uncertainty probing. Including these methods would provide a more comprehensive comparison with approaches that also leverage latent representations for uncertainty estimation.
2. The proposed method relies on the latent geometry of a pretrained classifier. However, since the backbone model is optimized purely for the classification objective, the learned feature space may suffer from feature collapse or reduced separability in certain regions, which could affect the reliability of the geometry-based uncertainty signals. This potential issue is not discussed or analyzed in the paper.
3. The experiments are conducted on several benchmarks, but large-scale graphs such as OGBN-Arxiv are not included. Evaluating the method on larger and more realistic graph datasets would help demonstrate its scalability and practical applicability.

---

> ### Author Rebuttal · Authors · 2026-03-31
>
> We thank the reviewer for the thoughtful feedback and recognition of our method's practicality, robustness, and clarity. We provide all supplementary materials (code, additional tables, t-SNE visualizations) in an [anonymous repository](https://anonymous.4open.science/r/X-EviProbe/).
>
> ## W1: Comparison with GPN and GEBM
>
> We note that X-EviProbe targets a fundamentally different setting from GPN: X-EviProbe performs UQ on a **frozen, pre-trained GNN**, while GPN trains an entirely new model with a built-in normalizing-flow density estimator and custom evidential loss. The two methods quantify uncertainty for **different underlying models**, making direct comparison less informative. Our primary baselines are post-hoc or lightweight methods operating on the same frozen backbone, which is the fair comparison setting.
>
> To address the reviewer's concern, we conducted additional experiments with both GPN and GEBM. **X-EviProbe maintains substantial leads across all datasets.** In OOD detection, GPN is runner-up only on AmazonPhoto and AmazonComputers, and GEBM only on PubMed, while X-EviProbe leads by +9.45%, +15.92%, and +30.3% respectively. In misclassification detection, GPN is runner-up only on Citeseer and AmazonComputers, yet X-EviProbe still leads by +6.34% and +6.18%. On all other datasets, neither GPN nor GEBM surpasses the existing baselines. Given that GPN and X-EviProbe do not even perform UQ on the same model, these large margins further validate our post-hoc approach. Full results are in our [anonymous repository](https://anonymous.4open.science/r/X-EviProbe/add_tables/).
>
> ## W2: Robustness Under Suboptimal Latent Geometry
>
> We address this from three angles.
>
> **The nearest-neighbor mechanism only requires local structure.** X-EviProbe does not assume globally well-separated clusters. The min operator in Eq. 4–5 depends only on the single closest reference point, making the method naturally tolerant to regions with reduced separability or multiple disjoint clusters. Our ablation (Table 4) confirms: replacing nearest-neighbor with Top-5/Top-10 averaging — which is more sensitive to global structure — consistently degrades performance (e.g., OOD AUROC drops from 98.19% to 91.18% on CiteSeer), as theoretically expected from Proposition 4.3.
>
> **Well-structured latent spaces sharpen the signal further.** When the classifier produces compact clusters (as commonly observed after convergence), the L1 distance (Eq. 4) becomes a sharper OOD indicator, and error anchors concentrate near decision boundaries, strengthening the aleatoric signal (Eq. 5). Our framework thus gracefully benefits from better representations while remaining robust to poor ones.
>
> **Experiments on severely distorted latent spaces.** We evaluate on three heterophilous benchmarks (Chameleon, Actor, Squirrel) where a standard GCN produces severely poorly-structured latent spaces (test accuracies as low as 19.5%). Despite this, X-EviProbe achieves the largest OOD improvements (+14.25%, +24.08%, +5.48% over the best comparable baselines). We also provide t-SNE visualizations (see tsne_visualizations/ in our anonymous repository) spanning well-structured to severely distorted latent spaces. Full heterophilous results are detailed in our response to Reviewer L6bW.
>
> ## W3: Large-Scale Graphs and Runtime
>
> We conducted experiments on **OGBN-Arxiv** (169K nodes, 1.2M edges, 40 classes with 15 held out as OOD), following EPN's exact protocol.
>
> **OOD Detection.** X-EviProbe achieves 89.61±1.87 AUROC, outperforming the best non-EPN baseline (Ensemble, 71.07) by +26.09% and EPN's own reported result (83.90) by +6.8% — without any hyperparameter tuning or OOD class exposure.
>
> **Misclassification Detection.** Our ablation (Appendix E.1, Table 7) reveals a consistent pattern: removing the logit component improves misclassification detection. With this task-specific configuration, X-EviProbe achieves 77.67±0.13, competitive with Ensemble (77.74±0.05).
>
> **Scalability.** A random subsample of 20K reference nodes (~12%) preserves strong performance: OOD AUROC 84.67 (still +19.1% above the best non-EPN baseline), misclassification AUROC 77.14. Further optimization via FAISS indexing can accelerate search to sublinear time.
>
> **Runtime (all on NVIDIA RTX 5090).** We decompose cost into extra training and inference overhead:
>
> |Dataset|Method|Extra Train|Inference|
> |---|---|---|---|
> |AmazonComp.|X-EviProbe|**0s**|1.11s|
> ||EGNN|6.51s|0.15s|
> ||EPN|4.04s|0.23s|
> ||SGNN-GKDE|15.47s|0.21s|
> ||Ensemble|76.74s|0.81s|
> ||MC Dropout|0s|1.09s|
> |OGBN-Arxiv|X-EviProbe (20K)|**0s**|5.16s|
> ||X-EviProbe (full)|**0s**|45.25s|
> ||Ensemble|835.63s|15.77s|
> ||MC Dropout|0s|45.32s|
> ||EPN|3.14s|4.16s|
>
> X-EviProbe with subsampling (5.16s) is faster than Ensemble (15.77s) and MC Dropout (45.32s), with zero extra training. Full runtime table is in add_tables/ of our anonymous repository.

---

> > ### Author Rebuttal · Reviewer_H8qS · 2026-04-04
> >
> > Thanks for the authors’ response. The updated results are very strong. Compared with the previous SOTA GEBM baseline, the proposed method outperforms it on all 7 datasets, with especially large gains on PubMed (+19.37), AmazonPhoto (+13.68), and AmazonComputers (+17.54) in AUROC. Averaged across the 7 datasets, the improvement is about 11 points, which is clearly substantial. Therefore, I raise my score accordingly.

---

> > > ### Author Response · Authors · 2026-04-04
> > >
> > > We sincerely thank the reviewer for the thorough re-evaluation and for recognizing the strength of our results. We are grateful for the constructive feedback throughout the review process, which has helped improve the paper.

---

### Decision · Program_Chairs · 2026-04-30

**Decision:**

Accept (regular)

**Comment:**

This paper proposes X-EviProbe, a practical, parameter-free, post-hoc uncertainty quantification framework for frozen graph neural networks that constructs Dirichlet evidence using latent distances and native outputs without retraining. Reviewers H8qS, nqow, i1dM, and L6bW unanimously praised the method's real-world utility, conceptual elegance, and strong empirical performance on out-of-distribution detection tasks. Initial reviews, however, raised shared concerns regarding the method's scalability to large graphs (H8qS), robustness to distorted latent spaces or diffuse boundaries (H8qS, L6bW), generalization beyond GCN backbones (i1dM), and computational runtime (nqow). During the discussion phase, the authors provided an exceptionally thorough rebuttal, executing extensive new experiments on the large-scale OGBN-Arxiv dataset, heterophilous benchmarks, and GAT architectures, while also providing exhaustive additive ablations and runtime comparisons. This rigorous response successfully resolved almost all criticisms, though Reviewer L6bW maintained a minor lingering reservation regarding misclassification efficacy on massive heterogeneous graphs compared to heavy ensemble methods, which the authors adequately addressed by pointing to their explicitly tailored 'w/o logits' configuration. I recommend acceptance because the method is technically sound, highly practical for immediate deployment, and backed by a remarkably robust rebuttal that thoroughly solidifies its claims. For the final manuscript, the authors should consider formally incorporate the newly provided large-scale OGBN-Arxiv evaluations, architectural generalizations, and detailed runtime analyses to ensure a comprehensive presentation of the framework's capabilities.